# Probing boron vacancy defects in hBN via single spin relaxometry

Alex L. Melendez [1], Ruotian Gong [2], Guanghui He [2], Yan Wang [3], Yueh-Chun Wu [4], Thomas Poirier [5], Steven Randolph [1], Sujoy Ghosh [1], Liangbo Liang [1], Stephen Jesse [1], An-Ping Li [1], Joshua T. Damron [6], Benjamin J. Lawrie [4], James H. Edgar [5], Ivan V. Vlassiouk [1], Chong Zu [2] & Huan Zhao [1] ✉

Spin defects in solids offer promising platforms for quantum sensing and memory due to their long coherence times and optical addressability. Here, we integrate a single nitrogen-vacancy (NV) center in diamond with scanning probe microscopy to detect, read out, and spatially map spin-based quantum sensors at the nanoscale. Using the boron vacancy ($V_B^-$) center in hexagonal boron nitride—an emerging two-dimensional spin system—as a model, we detect its electron spin resonance indirectly via changes in the spin relaxation time ($T_1$) of a nearby NV center, eliminating the need for optical excitation or fluorescence detection of the $V_B^-$. Cross-relaxation between NV and $V_B^-$ ensembles significantly reduces NV $T_1$, enabling quantitative nanoscale mapping of defect densities beyond the optical diffraction limit and clear resolution of hyperfine splitting in isotopically enriched $h^{10}B^{15}N$. Our method demonstrates interactions between spin sensors in 3D and 2D materials, establishing NV centers as versatile probes for characterizing otherwise inaccessible spin defects.

Spin-based quantum sensors leverage the coherent quantum states of localized electron or nuclear spins to detect minute variations in magnetic[1–3], electric[4,5], thermal[6–9], and strain fields[10] with remarkable sensitivity and nanoscale resolution[11,12]. Among the platforms developed to date, the nitrogen-vacancy (NV) center in diamond has emerged as a leading contender due to its long spin coherence times, room-temperature operation, and optical addressability[13–15]. The spin states of NV centers can be coherently manipulated using microwave fields and optically read out, enabling a wide range of applications including nanoscale magnetometry, quantum information processing, and biological imaging[16–21]. Despite its advantages, the NV center in bulk diamond faces significant limitations. In conventional architectures, NV centers typically reside tens of nanometers beneath the diamond surface, inherently limiting their proximity to external

sensing targets—a critical factor for achieving ultrahigh spatial resolution. Moreover, diamond's high refractive index results in significant total internal reflection, severely limiting optical fluorescence collection efficiency. These challenges have spurred the development of spin-based quantum sensors in atomically thin materials, wherein quantum defects reside directly at the surface, potentially enabling sub-nanometer proximity to the sensing environment and efficient photon extraction[22,23]. Currently, hexagonal boron nitride (hBN) is the only known two-dimensional material hosting optically active spin defects, making it uniquely promising for surface-proximal quantum sensing[24,25]. In addition to hBN with naturally occurring isotope ratios (hBN$_{nat}$), isotopically purified $h^{10}B^{15}N$ allows clear observation of the hyperfine structure, giving insight into electron-nuclear interactions as well as an increase in magnetic field sensitivity[26,27]. More broadly, the

[1]Center for Nanophase Materials Sciences, Oak Ridge National Laboratory, Oak Ridge, TN, USA. [2]Department of Physics, Washington University in St. Louis, St. Louis, MO, USA. [3]Computational Sciences and Engineering Division, Oak Ridge National Laboratory, Oak Ridge, TN, USA. [4]Materials Science and Technology Division, Oak Ridge National Laboratory, Oak Ridge, TN, USA. [5]Tim Taylor Department of Chemical Engineering, Kansas State University, Manhattan, KS, USA. [6]Chemical Sciences Division, Oak Ridge National Laboratory, Oak Ridge, TN, USA. ✉e-mail: zhaoh1@ornl.gov

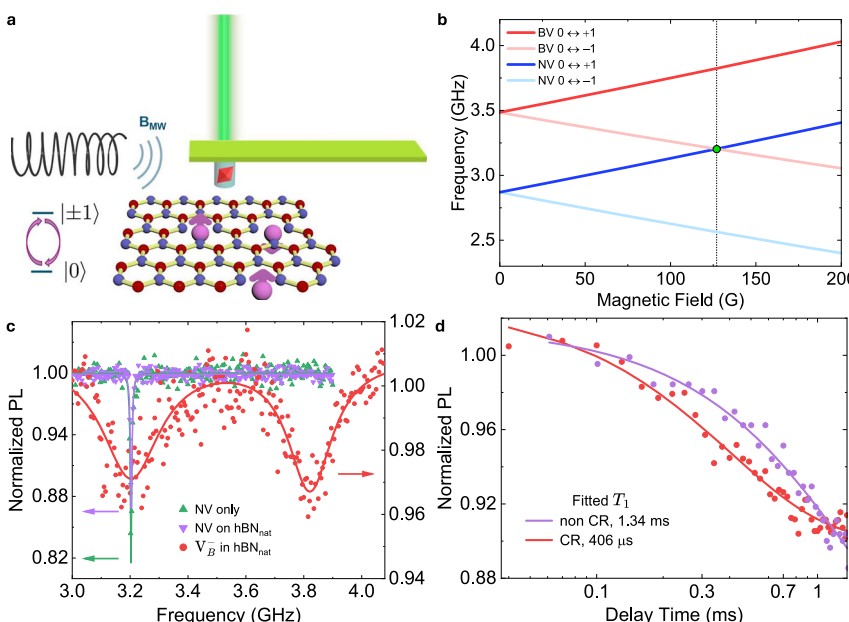

**Fig. 1 | NV $T_1$ relaxometry at CR condition with $V_B^-$ in hBN$_{nat}$. a** A scanning NV microscope containing a single NV center is brought near a 90–300-nm-thick hBN$_{nat}$ sample that has been irradiated with He ions to create an ensemble of $V_B^-$ centers. The NV center's photoluminescence (PL) is collected using a confocal optical setup. A microwave field $B_{MW}$ is supplied to both the NV and the nearest $V_B^-$ centers by an antenna positioned near the cantilever tip. When the microwave frequency matches the spin resonance frequency of the NV or $V_B^-$ centers, transitions between the ground state sublevels $m_s = 0 \leftrightarrow \pm1$ states are driven coherently. **b** Calculated spin resonance dispersions of the NV and $V_B^-$ centers with the magnetic field oriented 29° from the surface normal. Cross-relaxation occurs near 127 G bias, where the NV $m_s = 0 \leftrightarrow +1$ transition overlaps with the $V_B^-$ $m_s = 0 \leftrightarrow -1$ transition, highlighted by the green dot. **c** CW-ODMR spectra of the NV center before (green curve) and after (purple curve) engaging with the hBN sample, and of the $V_B^-$ centers in hBN (red curve), measured under the cross-relaxation condition shown in (**b**). All spectra are fitted using Lorentzian functions. The left and right vertical axes correspond to the NV and $V_B^-$ data, respectively. Only the $m_s = 0 \leftrightarrow +1$ transition of the NV is shown. The arrows denote, via their colors, which vertical axis each trace is plotted against. **d** NV spin $T_1$ measurement in non-CR condition (purple curve, fitted $T_1 = 1.34 \pm 0.19$ ms) and CR condition (red curve, $T_1 = 406 \pm 34$ µs), both measured after engaging the hBN$_{nat}$ sample. Curves are single exponential fits $\propto \exp(-t/T_1)$.

discovery and characterization of new spin-active defects in low-dimensional materials—some of which may ultimately emit in technologically relevant spectral ranges such as the near-infrared or telecom—requires experimental tools that do not rely on defect-specific optical readout. A key outstanding challenge is therefore the ability to directly image the spatial distribution, density, and charge state of spin-active defects at the nanoscale. Conventional optical and structural probes are typically diffraction-limited, thickness-averaged, or unable to distinguish spin-active from spin-inactive charge states, masking nanoscale inhomogeneities that govern decoherence, spin–spin interactions, and device functionality. These limitations motivate the development of a nanoscale, charge-state-selective spin-density imaging technique that is independent of the optical properties of the target spin.

In this work, we use the boron vacancy ($V_B^-$) center in hBN$_{nat}$ and h$^{10}$B$^{15}$N as a model system to demonstrate that NV-based $T_1$ relaxometry can read out and spatially map arbitrary spin-active quantum defects. The $V_B^-$ center is a point defect consisting of a missing boron atom in the hBN lattice, exhibiting a spin-1 ground state with optically detected magnetic resonance (ODMR) at room temperature[9,27–29]. By sweeping the magnetic field through the NV-$V_B^-$ spin cross-relaxation (CR) condition and monitoring the resulting changes in the NV center's spin relaxation time, we indirectly detect the boron vacancy's electron spin resonance (ESR) spectrum in hBN$_{nat}$—eliminating the need for direct optical excitation, microwave driving, or emission-based readout of $V_B^-$ defects. In addition, we demonstrate the ability of cross-relaxation spectroscopy to resolve the hyperfine structure in h$^{10}$B$^{15}$N, revealing nuclear spin information. Finally, we perform nanoscale spatial mapping of the cross-relaxation signal, revealing spatial heterogeneity in spin noise and quantitatively correlating this noise with

$V_B^-$ defect densities. Critically, our method addresses a key limitation of conventional defect-mapping techniques (e.g., Raman, electron microscopy) that cannot distinguish neutral from negatively charged boron vacancies, demonstrating that fewer than 10% of the vacancies are negatively charged and thus suitable as quantum sensors. These findings emphasize the versatility of scanning NV cross-relaxometry as a powerful method for probing emerging quantum spin systems, especially those with limited optical accessibility.

## Results

We investigated both chemical vapor deposition (CVD)-grown natural-abundance hBN (hBN$_{nat}$) and mechanically exfoliated isotopically enriched h$^{10}$B$^{15}$N (99% $^{10}$B, 99.5% $^{15}$N). The hBN$_{nat}$ membrane, with a nonuniform thickness ranging from 90 to 300 nm, was irradiated with helium ions to generate boron vacancy defects at an estimated density of ~0.1 defects per nm$^2$ per atomic layer (5400 ppm)[30,31]. Isotopically purified h$^{10}$B$^{15}$N crystals were synthesized via the atmospheric pressure high-temperature (APHT) method, neutron-irradiated, and subsequently exfoliated into flakes with an average thickness of ~50 nm. A single NV center, oriented along the diamond (100) plane and located approximately 9 ± 4 nm beneath the surface, was integrated into the apex of a tuning fork-based scanning-probe cantilever to form a scanning NV magnetometer. Operating in frequency-modulated contact mode with a finely tuned frequency offset, the best NV-to-sample distance was calibrated to be approximately 11.4 ± 1.5 nm (see Supplementary Note S1). Optical excitation and fluorescence collection from the NV center were achieved using a confocal microscope, while microwave fields for spin manipulation of both the NV and $V_B^-$ centers were delivered through an integrated radio-frequency (RF) antenna (Fig. 1a).

Figure 1b presents calculated ESR spectra for both the NV and $V_B^-$ centers under a magnetic field applied at an angle of 29° relative to the NV axis. At a bias field of 127 G, the NV $m_s = 0 \leftrightarrow +1$ and $V_B^-$ $m_s = 0 \leftrightarrow -1$ spin transitions become energetically degenerate at approximately 3.2 GHz. Under this resonance condition, the NV and $V_B^-$ spins can exchange energy non-radiatively via magnetic dipole-dipole coupling−a process known as cross-relaxation (CR)[32,33]−which results in an enhanced relaxation rate of the NV spin. Figure 1c shows continuous-wave ODMR (CW-ODMR) measurements of the NV center (only the $m_s = 0 \leftrightarrow +1$ transition is plotted) and the $V_B^-$ ensemble in hBN$_{nat}$ under the CR condition calculated in Fig. 1b, confirming spectral overlap of their respective spin transitions. Compared to the single NV center, the $V_B^-$ ensemble ODMR spectrum exhibits significantly broader linewidths and lower contrast, limiting its performance as a standalone quantum sensor. Upon bringing the NV sensor into contact with the hBN sample, the ODMR contrast of the NV $m_s = 0 \leftrightarrow +1$ transition decreases from 18.5% to 13.1%. This reduction in contrast may reflect faster spin population equalization, i.e., a shorter $T_1$, though other processes such as charge dynamics may also explain this decrease[34]. The $T_1$ reduction is confirmed by full $T_1$ measurements, as shown in Figure 1d. While engaged with the sample, a magnetic field is applied to bring the NV and $V_B^-$ into the CR condition, decreasing the NV spin $T_1$ from $1.34 \pm 0.19$ ms to $406 \pm 34$ μs.

The total relaxation rate $\Gamma_1 = 1/T_1$ of the NV center can here be modeled as a contribution from three terms[35]:

$$\Gamma_1 = \Gamma_1^{int} + \Gamma_1^{bath} + \Gamma_1^{CR}, \qquad (1)$$

where $\Gamma_1^{int}$ is the intrinsic relaxation rate of the NV, $\Gamma_1^{bath}$ is the contribution from the ambient spin noise produced by the hBN, and $\Gamma_1^{CR}$ is the contribution due to the CR condition being met between the NV and $V_B^-$ spins. Taking dipole-dipole interaction between the $V_B^-$ spin bath and the NV as the dominant interaction, the ensemble-averaged relaxation rate is[36,37]

$$\Gamma_1^{CR} = \left\langle \left( \frac{J_0 \mathcal{A}_j}{r_j^3} \right)^2 \frac{2}{\gamma + \Gamma_2^{NV} + \Gamma_2^{V_B^-}} \right\rangle, \qquad (2)$$

contains the angular dependence between the NV and $j^{th}$ $V_B^-$ spin, $\Gamma_2^{NV}$ and $\Gamma_2^{V_B^-}$ are the NV and $V_B^-$ transition linewidths, and $\gamma$ is the interaction-induced intrinsic linewidth. Using Eq. (2), a Monte Carlo simulation of the NV-$V_B^-$ dipolar interactions was performed, showing that the cross-relaxation rate of $\Gamma_1^{CR} = 1.72$ kHz seen in Fig. 1d corresponds to a $V_B^-$ density of about 220 ppm (see Supplementary Note S4 for details).

Hyperfine splitting in ODMR spectra arises from the magnetic interaction between an electron spin and nearby nuclear spins, leading to multiple resonance lines that reflect the local nuclear spin environment. These splittings are essential for nuclear-spin-based sensing, as they enable identification, control, and readout of individual nuclear species, allowing for nanoscale magnetic resonance and enhanced spectral selectivity in quantum sensing applications. Surrounding the $V_B^-$ electron spin are three nearest-neighbor nitrogen nuclei, whose hyperfine interactions split the ODMR transitions. In natural-abundance hBN (hBN$_{nat}$), the dominant isotope is $^{14}$N ($I = 1$), which results in seven allowed hyperfine transitions. These lines overlap and broaden due to nuclear spin mixing and inhomogeneities, producing the broad ODMR spectra observed in Fig. 1c. In contrast, isotopically enriched h$^{10}$B$^{15}$N contains $^{15}$N nuclei ($I = 1/2$), yielding four hyperfine-split lines corresponding to the total nuclear spin projection $m_I = \{ -3/2, -1/2, +1/2, +3/2 \}$[26].

Figure 2a shows a representative CW-ODMR spectrum of the $V_B^-$ $m_s = 0 \leftrightarrow -1$ transition in h$^{10}$B$^{15}$N, exhibiting clearly resolved hyperfine structure. In total, 24 CW-ODMR spectra were recorded for the NV $m_s = 0 \leftrightarrow +1$ and 12 for the $V_B^-$ $m_s = 0 \leftrightarrow -1$ transitions as a function of an out-of-plane (OOP) magnetic field, with the extracted center frequencies plotted in Fig. 2b. The average hyperfine splitting between the $m_I = \pm 1/2$ levels is $|A_{zz}| = 2\pi \times (67 \pm 2)$ MHz (see Supplementary Note S5 for additional data). Using these values, the positions of the four hyperfine transitions (dashed lines) are plotted, from which four distinct cross-relaxation points with the NV $m_s = 0 \leftrightarrow +1$ transition are identified at 101, 114, 127, and 140 G.

To rapidly assess the magnetic field dependence of NV spin relaxation, we employed a single-$\tau$ $T_1$ relaxometry technique. In this method, the NV center is first polarized into the $m_s = 0$ spin state, followed by a fixed evolution time $\tau$, after which the photoluminescence (PL) intensity is recorded. A reduced ratio PL($\tau$)/PL(0) indicates enhanced relaxation and thus a shorter $T_1$ time. Figure 2c presents PL($\tau$)/PL(0) as a function of OOP magnetic field after engaging the NV tip with an exfoliated h$^{10}$B$^{15}$N flake, revealing four dips corresponding to the hyperfine transitions of $V_B^-$ centers in h$^{10}$B$^{15}$N. These features occur when the NV ESR frequency matches the hyperfine-split ESR frequencies of the h$^{10}$B$^{15}$N, confirming that our microwave-free cross-relaxation method can probe both electron and nuclear spin properties and recover the spin resonance spectra.

For comparison, we performed the same field-dependent PL($\tau$)/PL(0) measurement after landing an NV tip on a hBN$_{nat}$ sample. As shown in Fig. 2d, the resulting curve closely resembles the CW-ODMR spectrum of $V_B^-$ in hBN$_{nat}$ given in Fig. 1c. Compared to the $V_B^-$ ODMR spectrum, the field-dependent single-$\tau$ $T_1$ relaxometry curve−hereafter referred to as the $T_1$-magnetic resonance ($T_1$-MR) curve−exhibits enhanced signal contrast, limited primarily by the intrinsic spin-dependent fluorescence contrast of the NV center. Additional $T_1$-MR data acquired on the same sample using an independent NV probe are presented in Supplementary Note 7, demonstrating that the measured $T_1$-MR response is reproducible across different tips.

Diamond tips with long NV $T_1$ times offer higher sensitivity but require significantly longer acquisition times to record a complete $T_1$ decay curve, making single-$\tau$ $T_1$-MR a more realistic approach for field-dependent measurements. However, the PL($\tau$)/PL(0) signal is also influenced by the NV readout contrast, which in our experiments decreases approximately linearly with increasing magnetic field (Supplementary Fig. S21). This introduces a smooth, non-resonant background in single-$\tau$ $T_1$-MR traces that is unrelated to cross-relaxation. To remove this contribution, each $T_1$-MR curve in Fig. 2 was baseline-corrected by fitting the non-resonant background to a linear function and subtracting it, thereby isolating the cross-relaxation features. A derivation and full description of this baseline-correction procedure are provided in Supplementary Note 5.

NV tips with shorter $T_1$ times, although less sensitive, allow acquisition of full $T_1$ decay curves at each field within a reasonable timescale, thereby enabling direct extraction of $T_1$ values and eliminating the influence of field-dependent ODMR contrast. All prior measurements presented in this study were performed with tips having $T_1 > 1$ ms to maximize sensitivity (see Supplementary Information Table S1 for a list of NV tips used). In contrast, Figure 2e shows relaxation from a tip with a significantly shorter $T_1$ of 191 μs. Once the tip engages with a h$^{10}$B$^{15}$N flake, full $T_1$ relaxation curves are acquired at varying magnetic fields, and the corresponding $T_1$ values are extracted. Figure 2f displays the $T_1$-field dependence, clearly resolving the hyperfine structure of h$^{10}$B$^{15}$N. We note that the relative dip depths in $T_1$-MR can deviate from the hyperfine intensity ratios observed in direct ODMR, particularly for short-$T_1$ probes where broadened response and partial saturation/nonlinear cross-relaxation can equalize the apparent contrasts. Importantly, these effects do not affect the extracted resonance frequencies or line-widths, which underpin the analysis.

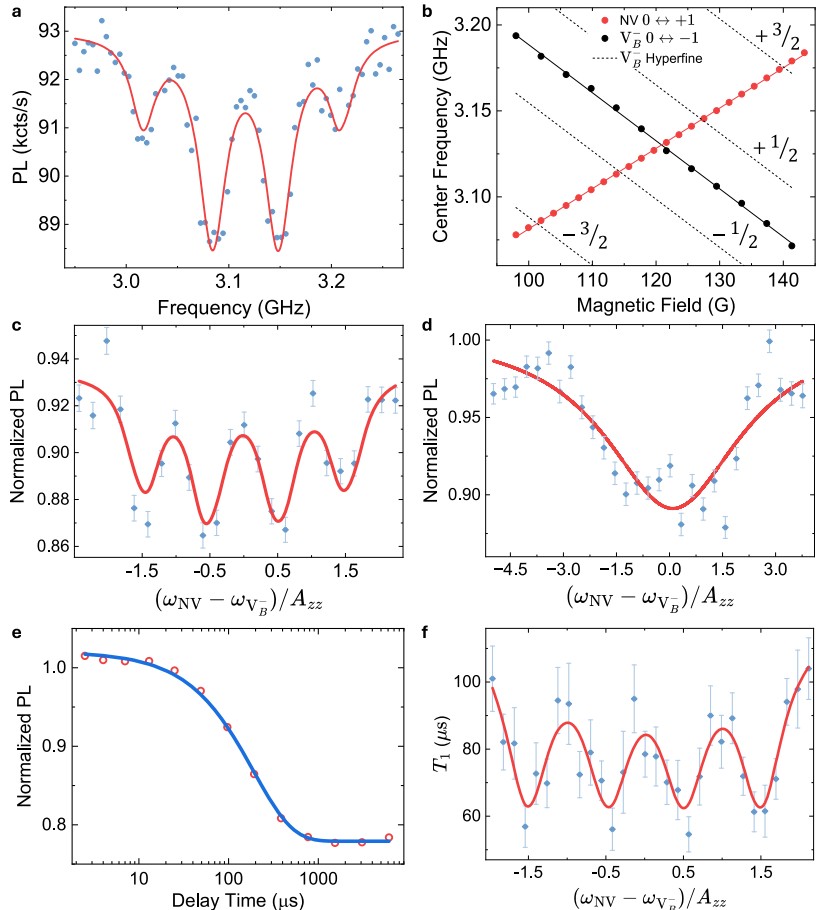

**Fig. 2 | $T_1$-MR detection of hyperfine structure. a** CW-ODMR spectrum of $V_B^-$ centers in $h^{10}B^{15}N$ showing four hyperfine-split lines, fitted with Lorentzian functions (red). A magnetic field of 123 G is applied out-of-plane. **b** Extracted ODMR center frequencies of the NV center (red) and $V_B^-$ centers (black) as a function of magnetic field, with solid lines representing fitted curves. The hyperfine line positions (dashed) are calculated using the average $A_{zz}$, revealing four expected cross-relaxation conditions corresponding to the crossover points between the NV and $V_B^-$ transitions. **c** NV single-$\tau$ $T_1$-MR detection of $V_B^-$ hyperfine structure by sweeping the magnetic field through the four CR conditions. The magnetic field is converted to frequency detuning between the NV and $V_B^-$ transitions and plotted in units of $A_{zz}$ in $h^{10}B^{15}N$ (67 MHz). The data are fitted using a sum of four Lorentzian functions, where dips 1 and 3 share one set of parameters, and dips 2 and 4 share another (fit shown in red). Here, the free-evolution time $\tau$ is chosen to be 2 ms, given that the

tip's non-CR $T_1$ measured immediately before this $T_1$-MR measurement is $2.09 \pm 0.25$ ms (see Supplementary Fig. S23a). **d** NV single-$\tau$ $T_1$-MR measurement of $hBN_{nat}$ under varying magnetic field. The field axis is converted to frequency detuning and plotted in units of $A_{zz}$ in $hBN_{nat}$ (44 MHz). Data are fitted using a Lorentzian function (red curve). The $\tau$ in this case is 0.7 ms, and the tip's $T_1$ measured at the non-CR condition is $1.21 \pm 0.32$ ms (see Supplementary Fig. S23b). Error bars of (**c**) and (**d**) represent the normalized shot noise level. **e** A short-$T_1$ NV tip's spin $T_1$ relaxation curve before engaging the sample, showing $T_1$ of $191 \pm 7$ μs. **f** NV $T_1$ values measured as a function of magnetic field (blue symbols with error bars), showing four distinct cross-relaxation dips. The curve is fitted with a sum of Lorentzian functions and plotted in units of $A_{zz}$ in $h^{10}B^{15}N$ (67 MHz). Error bars represent $1\sigma$ uncertainties of fitted $T_1$ values.

Understanding the spatial distribution of spin defects is essential not only for optimizing defect creation and implantation processes but also for probing nanoscale spin–spin interactions that govern decoherence and quantum entanglement. Raman spectroscopy, particularly through mapping of defect-associated vibrational modes, has long served as a valuable technique for visualizing defect distributions[38–41]. However, its diffraction-limited spatial resolution (~500 nm) restricts access to nanoscale variations that are often critical in quantum materials. More importantly, most conventional defect-mapping techniques—including Raman and transmission electron microscopy (TEM)—cannot distinguish between neutral ($V_B^0$) and negatively charged ($V_B^-$) boron vacancy centers. Since only $V_B^-$ possesses the desired spin properties and optical addressability, it is crucial to develop spatial mapping methods that selectively target $V_B^-$ defects. Using the NV center's spin $T_1$ as imaging contrast, scanning NV microscopy offers both high spatial resolution and selective

sensitivity to $V_B^-$ centers. When utilizing shallowly implanted NV centers, various sensing techniques have demonstrated spatial resolution down to ~10 nm, enabling direct imaging of magnetic noise from paramagnetic spin distributions with nanometric precision[42,43]. Under cross-relaxation conditions, regions of the hBN sample with higher surface $V_B^-$ densities generate stronger magnetic noise at the NV resonance frequency, leading to enhanced NV spin relaxation. By raster-scanning the NV sensor and performing single-$\tau$ $T_1$ relaxometry at each pixel, a high-resolution spatial map of $V_B^-$ density can be constructed, revealing nanoscale variations in defect distribution well below the diffraction limit.

Figure 3a shows the optical image of a $hBN_{nat}$ flake after helium ion irradiation, while Fig. 3b presents the corresponding Raman intensity map of the $E'/E_{2g}$ vibrational peaks. The $E_{2g}$ mode, associated with in-plane vibrations of $sp^2$-bonded boron and nitrogen atoms, is characteristic of pristine hBN[44]. In contrast, the $E'$ peak at ~1290 cm$^{-1}$

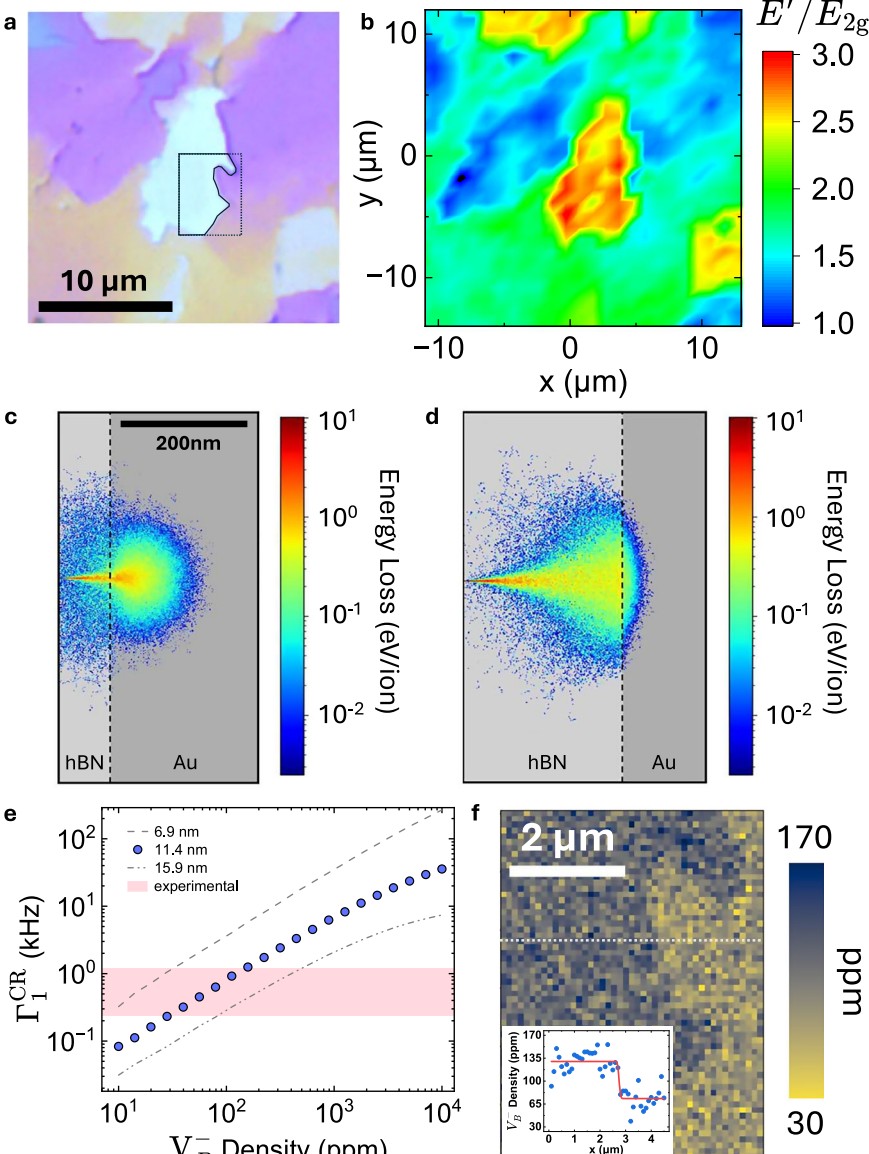

**Fig. 3 | Nanoscale imaging of $V_B^-$ density in hBN$_{nat}$ using NV-$V_B^-$ cross-relaxometry. a** Optical microscope image of hBN$_{nat}$ sample on a gold surface, with different colors corresponding to variations in the hBN thickness. The light-colored area in the center has a thickness of ~90 nm compared to the surrounding pink/purple areas with a thickness of ~190 nm. Here, a portion of the right edge of the light-colored area is highlighted in a blue polygon, which was measured by the scanning NV in (**f**). **b** Raman spectroscopy map showing the ratio of the $E'$ peak to the $E_{2g}$ peak across the sample area displayed in (**a**). **c** SRIM simulations of the ion energy losses (normalized per ion) of 30 keV He$^+$ (10,000 ions simulated) for 90 nm and (**d**) 250 nm hBN on gold, suggesting thickness-dependent nonuniform defect

density distribution. **e** Monte Carlo simulation of the cross-relaxometry contribution to the NV relaxation rate $\Gamma_1^{CR}$ as a function of the $V_B^-$ density and three different NV-to-sample distances. The horizontal gray bar shows the values of $\Gamma_1^{CR}$ associated with the range of the color scale in (**f**). **f** $V_B^-$ density map, obtained as a single-$\tau$ $T_1$ spatial scan over a free-evolution time $\tau = 250$ μs, which was converted to $V_B^-$ density via the Monte Carlo simulation in (**e**). The size of each pixel is 100 nm. The measurement was conducted at the CR condition. Inset: The profile of $V_B^-$ density as a function of position, over the line indicated by the white dashed line in (**f**). The data is fit to the edge-spread function (red), yielding a step width of 46 nm.

emerges following helium ion irradiation and is attributed to the formation of $V_B^-$ defects[31,45]. Stopping and Range of Ions in Matter (SRIM) simulations were performed to estimate the He$^+$ energy loss for 90-nm-vs 250-nm-thick hBN on a gold substrate, showing thickness-dependent ion backscattering (Fig. 3c, d and Supplementary Note S6). Based on these simulations, it is expected that a higher proportion of defects is concentrated at the surface of the 90 nm film and delocalized from the impact point. Conversely, it would be expected that defects in the 250 nm film would remain localized in the near-surface region, but broaden significantly throughout the thickness of the film. Hence, regions of varying hBN thickness—visible as distinct colors in the optical image—exhibit different $V_B^-$ defect

densities[30]. Consequently, structural features observed in both the optical and Raman maps are spatially correlated.

To further resolve local spin defect distributions beyond the optical diffraction limit, we performed a single-$\tau$ $T_1$ relaxometry scan over the region of interest with a 100 nm step size. Figure 3e shows Monte Carlo simulations of the cross-relaxation rate Eq. (2) at various $V_B^-$ densities, allowing conversion between a spatial map of NV relaxation to that of $V_B^-$ density depicted in Fig. 3f (see Supplementary Note S4). This NV-based nanoscale mapping resolves spatial features in the defect density that are not discernible in the Raman data, including abrupt grain boundaries in the CVD-grown hBN. Additional correlative imaging of the same sample region is provided in Supplementary

Note S9. In addition, as a quantitative measurement, it offers the spin-density value at each pixel. Notably, the spatial resolution of the scanning NV relaxometry measurement and thereby the $V_B^-$ density is primarily limited by NV-sample distance and can be improved to ~10 nm by reducing the scan step size, although this significantly increases the total acquisition time.

## Discussion

The shot-noise-limited magnetic sensitivity of a spin-1 quantum sensor operating under the CW-ODMR protocol is given by[46]:

$$\eta_{CW} \propto \frac{\Delta\nu}{C_{CW}\sqrt{R}}, \tag{3}$$

where $\Delta\nu$ is the ODMR linewidth, $C_{CW}$ is the ODMR contrast, and $R$ is the photon detection rate. Optimal magnetic field sensitivity, therefore, requires a narrow ODMR linewidth, high contrast, and efficient photon collection. In practice, the ODMR contrast of $V_B^-$ ensembles is typically low (<5%) for several reasons: (1) the hBN contains a fraction of bright defects that do not exhibit ODMR; (2) the spin-lattice relaxation time $T_1$ is typically short (<10 μs), which limits spin polarization fidelity; and (3) the spin-dependent fluorescence contrast is intrinsically weak due to competing non-radiative decay pathways via intersystem crossing. In contrast, our $T_1$-MR measurement, which reproduces the ODMR spectral profile, yields higher contrast and does not rely on the optical readout of the target spin species. The method detects population decay from the optically polarized NV spin $m_s = 0$ state toward thermal equilibrium. Assuming a 30% intrinsic NV ODMR contrast, the expected change in PL from the polarized state to the thermalized population yields a theoretical PL contrast of ~20% in all-optical $T_1$ relaxometry. When performing field-dependent full $T_1$ measurements and using the extracted $T_1$ values—rather than the PL($\tau$)/PL(0) ratio—as the sensing metric, the $T_1$-MR contrast of NV−$V_B^-$ spin resonance detection can be enhanced by an order of magnitude. For instance, as shown in Fig. 1d, under cross-relaxation conditions, the NV $T_1$ decreases from 1.34 ms to 406 μs, corresponding to a contrast of approximately 70%. The field-dependent full $T_1$ measurements using a short-$T_1$ NV tip, shown in Fig. 2f, reveal the hyperfine structure of h$^{10}$B$^{15}$N with approximately 50% $T_1$-MR contrast.

Another key advantage lies in the photon detection efficiency. The fluorescence spectrum of $V_B^-$ peaks at ~850 nm, whereas silicon avalanche photodiodes (APDs) are most efficient near 650 nm, leading to sub-optimal detection. In contrast, our NV-based relaxometry approach relies on the NV center's own fluorescence for readout, which allows for the detection of spin species across a broad spectral range. This includes optically dark defects as well as those emitting at technologically important telecom wavelengths, all using a single, low-cost APD. For spin defects that emit within the same spectral range as NV centers—and therefore cannot be isolated using conventional optical filters—we developed a dark readout method that enables effective separation through time-gated detection, exploiting differences in fluorescence dynamics rather than spectral properties (see Supplementary Note S2). We acknowledge that a limitation of relaxometry is its longer acquisition time due to its pulsed nature—typically about 3000 times longer than that of CW-ODMR. This corresponds to a sensitivity degradation of approximately $1/\sqrt{3000}$, or about 55 times lower sensitivity, assuming equal acquisition time. However, this limitation can be offset by the enhancement in signal contrast and further mitigated by using dense NV ensembles, which significantly improve the signal-to-noise ratio and reduce integration time—paving the way for scalable, high-contrast, and broadband quantum sensing.

Supplementary Note 3 presents a quantitative model for single-$\tau$ $T_1$-MR measurements and derives the optimal wait time $\tau$ that maximizes signal-to-noise by balancing contrast against photon shot noise and measurement duty cycle. Within the same framework, we evaluate the shot-noise-limited magnetic field sensitivity $\eta(\tau)$ near the NV−$V_B^-$

cross-relaxation condition. This analysis further indicates that $T_1$-MR readout using the bright NV fluorescence can outperform direct PL-based ODMR of $V_B^-$ centers in the few-spin (or nanoscale-cluster) limit. In this regime, the $V_B^-$ PL from a ~1–10 nm defect cluster can be far below the level typically detected in confocal measurements that collect fluorescence from a diffraction-limited volume (~500 nm lateral diameter). By contrast, scanning NV cross-relaxometry can place the NV directly above the cluster, and the cross-relaxation contribution to the NV relaxation rate is dominated by the nearest defects and decays steeply with separation ($\propto r^{-6}$), enabling a measurable $T_1$-MR signal even when $V_B^-$ ODMR is not feasible.

Under helium ion irradiation at a dose of 50 He$^+$/nm$^2$, scanning NV cross-relaxometry reveals a $V_B^-$ density of 30–170 ppm, which is substantially lower than the ~5400 ppm total vacancy density measured by atomic-resolution TEM imaging[31]. This indicates that most boron vacancies remain in the neutral $V_B^0$ state. Our inferred negatively charged fraction of 3.1–9.3% aligns with the 1–10% $V_B^-$ density reported by a previous work[47]. Traditional optical methods average over regions that include both high and low defect densities, which can obscure sharp local variations. In contrast, our nanoscale relaxometry technique offers precise, spatially resolved measurements that preserve local contrast and reveal fine-scale features. In addition, unlike optical methods that integrate signals over the entire sample thickness, our technique primarily probes surface defect densities—precisely the region most critical for quantum sensing applications (see Supplementary Note 4 and Fig. S17). A major limitation for many optical and electrical probes is that they cannot effectively distinguish $V_B^-$ from $V_B^0$ or isolate the near-surface defect layer relevant for device operation. By tuning the magnetic field to the NV−$V_B^-$ cross-relaxation condition, our NV-based $T_1$ relaxometry becomes selectively sensitive to spin-active $V_B^-$ defects whose ESR matches that of the NV, while neutral $V_B^0$ and other ESR-detuned centers contribute negligibly to the contrast. As a proof-of-principle relevant to device operation, we combine electrostatic gating with NV−$V_B^-$ cross-relaxometry (Supplementary Note 8) by tracking the gate-dependent change in NV $T_1^{-1}$ at the cross-relaxation field, which directly reports the surface-proximal $V_B^-$ spin density. We observe a ~30% modulation, consistent with our surface band-bending hypothesis, in which electrostatic gating induces a band bending that alters the local defect charge-state population in a shallow, surface-proximal region of hBN (see Supplementary Note 8 for more discussion). This modulation is substantially larger than the few-percent changes inferred from thickness-averaged photoluminescence[48], underscoring the charge-state selectivity and surface sensitivity of our non-invasive approach.

Previous studies have demonstrated cross-relaxation (CR) between NV centers and nearby proton spins or other spin defects within diamond, and the ability of NV centers to read out other electron spins has been established in several pioneering works[32,33,49–52]. However, these experiments mainly rely on dynamical-decoupling protocols, which require (i) microwave driving of the NV center, (ii) microwave or RF driving of the target spin, (iii) prior knowledge of the target spin resonance (frequency and $\pi$-pulse length), and (iv) nanosecond-scale timing control; in practice, such pulsed schemes are most convenient for frequencies up to the hundreds-of-MHz range, where high-power, short $\pi$ pulses are experimentally tractable. Cross-relaxation between an NV and spin-based quantum sensors in external materials—especially those of different dimensionalities—has not yet been achieved, and demonstrating such interactions is critical for extending NV-based detection to a broader range of quantum systems. Prior CR experiments have also relied on static diamond slabs, where the spatial relationship between the NV and target spins is fixed, limiting control over interaction strength and spatial resolution. In contrast, our method uses NV-$V_B^-$ cross-relaxation $T_1$ relaxometry as a microwave-free, all-optical readout: the protocol (i) does not require any microwave

control of the NV center, (ii) only requires microsecond–millisecond timing resolution, (iii) operates with minimal prior knowledge of the target spin, and (iv) yields an ESR spectrum at GHz frequencies set by the applied magnetic field. Furthermore, by combining this cross-relaxation contrast with quantitative modeling, we can not only detect the presence of target spins but also map their local spin density at the nanoscale. Implemented in a scanning NV geometry, this approach provides two major advantages: precise nanoscale control over the NV-target distance to optimize dipole-dipole coupling, and the capability to spatially map spin noise with sub-diffraction resolution. This platform also enables simultaneous manipulation and readout of spin dynamics—achieved by adjusting the NV-target separation or applying localized strain to the target using the same diamond tip employed for CR-based sensing.

In conclusion, we demonstrate NV-based spin relaxometry as a versatile method for characterizing and spatially mapping spin-active quantum defects, exemplified by boron vacancy defects in 2D hBN. Leveraging the established platform of scanning NV microscopy, our technique bypasses the conventional requirement for dedicated optical excitation and detection infrastructure for each new defect species. This enables all-optical relaxometry-based ESR measurements with high spectral contrast and nanoscale spatial precision. Our approach streamlines and standardizes the discovery of novel spin-active defects, including those emitting at telecom wavelengths or optically dim defects that are otherwise inaccessible with low-cost detectors. Furthermore, our approach enables heterogeneous quantum architectures by decoupling sensing and readout functionalities into distinct qubits and thus harnessing their complementary strengths while minimizing individual limitations. Building upon this demonstration, we envisage future directions that include controlled gate operations between NV centers and nearby two-dimensional spin defects, entanglement-enhanced quantum sensing, and scanning-probe NMR spectroscopy via cross-relaxation protocols.

During revision of this manuscript, we became aware of a related study published by Sun et al.[53] reporting nanoscale imaging of spin defects in two-dimensional materials. Their results are complementary to those reported here.

## Methods
### Sample preparation
Flakes of thick hBN$_{nat}$ were synthesized by chemical vapor deposition (CVD) using solid boron precursors in a nitrogen atmosphere. A flake of variable thickness (90–300 nm) was transferred onto a gold coplanar waveguide (CPW), which was lithographically patterned onto a sapphire substrate. However, this CPW was not used as a waveguide for microwaves in the course of this work. The hBN$_{nat}$ was irradiated with a beam of He ions at a dose of 50 He$^+$/nm$^2$ at an energy of 30 keV using a Zeiss Orion NanoFab Helium ion microscope. At this energy, the penetration depth in hBN was about 240 nm. As a result, more He ions reached the gold where the hBN$_{nat}$ was thinner, which increased the production of V$_B^-$ centers.

The h$^{10}$B$^{15}$N crystal flakes were grown by the atmospheric pressure high-temperature (APHT) method[54,55]. Briefly, the process starts by mixing high-purity elemental boron, enriched in the $^{10}$B isotope to 99%, with nickel and chromium with weight ratios of 4:48:48, respectively. The mixture is then heated under a blanket of nitrogen enriched in the $^{15}$N isotope to 99.5% at a pressure of 850 torr to 1550°C, to produce a homogeneous molten solution. After 24 h, the solution is slowly cooled at 4°C/h, to 1500°C, then at 50°C/h to 1300°C, and 200°C/h to room temperature. The h$^{10}$B$^{15}$N solubility is decreased as the temperature is reduced, causing crystals to precipitate on the surface of the metal. The h$^{10}$B$^{15}$N flakes were exfoliated from the metal with thermal release tape. The free-standing h$^{10}$B$^{15}$N crystalline flakes were typically 10–20 μm thick. To create boron vacancies, the h$^{10}$B$^{15}$N flakes were neutron-irradiated. Samples were

irradiated in The Ohio State University nuclear reactor, operated at a power of 300 kW for 1 h, corresponding to neutron fluences of $1.4 \times 10^{16}$ neutron/cm$^2$. Prior to the cross-relaxometry experiments, micrometer-scale flakes were exfoliated from irradiated macroscopic crystals using standard mechanical exfoliation. This process yielded flakes with thicknesses of approximately 50 nm, used for the measurements shown in Fig. 2, and approximately 15 nm, used for the data presented in Fig. S26.

Photoluminescence (PL) and Raman spectroscopy were performed on hBN$_{nat}$ using a commercial InVia Qontor confocal system (Renishaw) to characterize defect implantation. The sample was excited using a 532 nm laser delivered through a 100 × Leica objective (NA = 0.85). Scattered signals were collected by the same objective, spectrally filtered using ultra-narrow notch filters, and subsequently dispersed onto diffraction gratings with groove densities of 1800 lines mm$^{-1}$ (Raman measurements) and 300 lines mm$^{-1}$ (PL measurements). The PL spectrum of the h$^{10}$B$^{15}$N samples was not measured but is expected to have a similar spectrum reported in previous studies[27].

### Experimental setup
NV measurements were performed with a commercial Qnami ProteusQ™ room-temperature scanning NV microscope. A cantilever with a diamond nanopillar containing a single NV center was brought near (<5 nm) to the hBN surface. The NV center was excited using a 520 nm laser (~20 μW), and the photoluminescence was collected via a confocal optical setup. Microwaves (~0.1 W input) were supplied by a nearby shorted coaxial cable brought within ~15 μm of the NV, using a Qnami MicrowaveQ signal generator or a Keysight M8195A Arbitrary Waveform Generator.

### Sensing protocols
Continuous-wave (CW) optically detected magnetic resonance (ODMR) measurements of the NV center and the V$_B^-$ ensemble, as shown in Figs. 1c and 3, were performed under continuous laser excitation and microwave drive. A static magnetic field was applied using a neodymium ring magnet mounted on a three-axis translation stage, while the microwave frequency was swept. To suppress fluorescence from the V$_B^-$ ensemble, a 750 nm short-pass filter was used during NV ODMR measurements.

Pulsed NV $T_1$ relaxometry measurements were conducted using a 3 μs laser pulse to initialize the NV spin into the $m_s = 0$ state. The spin was then allowed to relax toward thermal equilibrium during a dark interval $\tau$, during which microwave excitation was applied. Relaxation was detected as a reduction in photoluminescence (PL) during a subsequent readout sequence, preceding the next initialization pulse. At each readout pulse sequence, a reference PL signal, PL(0), is recorded after re-polarizing the spin into the $m_s = 0$ state. By varying $\tau$, exponential decay curves $\Delta PL(\tau) = PL(\tau)/PL(0) \propto \exp(-\tau/T_1)$, such as those shown in Fig. 1d were obtained and fit, from which the $T_1$ value was extracted.

To reduce acquisition time, iso-$T_1$ relaxometry measurements were performed by fixing time $\tau$. In this case, the relative spin relaxation signal was estimated as $\Delta PL = \frac{PL(\tau)}{PL(0)}$, providing a rapid proxy for magnetic noise mapping with reduced temporal overhead.

## Data availability
The raw data of the main text figures are available in the Zenodo database[56]. Further data are available from the corresponding author upon request.

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

## Acknowledgements

The scanning NV microscopy, hBN_nat synthesis, and nanofabrication were supported by the Center for Nanophase Materials Sciences (CNMS), which is a US Department of Energy, Office of Science User Facility at Oak Ridge National Laboratory. Spin relaxation measurements were supported by the Laboratory Directed Research and Development Program of Oak Ridge National Laboratory, managed by UT-Battelle, LLC, for the U.S. Department of Energy. The RF controls were supported by the U.S. Department of Energy, Office of Science, Basic Energy Sciences, Materials Sciences and Engineering Division. Support for $h^{10}B^{15}N$ crystal growth was provided by the Office of Naval Research, award number N00014-22-1-2582. Neutron irradiation of the $h^{10}B^{15}N$ crystals was supported by the U.S. Department of Energy, Office of Nuclear Energy, under DOE Idaho Operations Office Contract DE-AC07-051D13417 as part of a Nuclear Science User Facilities experiment. This manuscript has been authored by UT-Battelle, LLC, under contract DE-AC05-00OR22725 with the US Department of Energy (DOE). The US government retains, and the publisher, by accepting the article for publication, acknowledges that the US government retains a non-exclusive, paid-up, irrevocable, worldwide license to publish or reproduce the published form of this manuscript, or allow others to do so, for US government purposes. DOE will provide public access to these results of federally sponsored research in accordance with the DOE Public Access Plan (http://energy.gov/downloads/doe-public-access-plan). R.G., G.H. and C.Z. acknowledge support from the National Science Foundation under grant No. 2514391.

## Author contributions

H.Z. conceived the project and performed the NV relaxometry experiments. A.L.M. and H.Z. analyzed the data and co-wrote the manuscript with input from all other authors. A.L.M., Y.W., R.G., and G.H. developed the theoretical model, with additional input from L.L., C.Z., and J.T.D. R.G., G.H., and C.Z. performed the Monte Carlo simulation and quantified the defect density. I.V.V. synthesized the hBN_nat samples and performed optical spectroscopy. T.P. and J.H.E. synthesized the $h^{10}B^{15}N$ samples and organized the neutron irradiation. S.R. conducted the helium ion implantation. S.G. fabricated the coplanar waveguide. Y.C.W. and B.J.L. assisted with the microwave delivery setup. A.-P.L., S.J., and B.J.L. provided technical support for the scanning NV microscope measurements.

## Competing interests

The authors declare no competing interests.
