## [Transparent Peer Review file · Nature Communications]

Probing Boron Vacancy Defects in hBN via Single Spin Relaxometry

Corresponding Author: Dr Huan Zhao

Version 1:

Reviewer comments:

Reviewer #1

(Remarks to the Author)

The manuscript titled "Probing Spin Defects via Single Spin Relaxometry" presents a novel study that utilizes scanning NV centers to read out the quantum states of VB^- and map its distribution. In this way, the low spin contrast of VB^- can be circumvented by relying on the high spin contrast and NV. Meanwhile, other spin-active quantum defects in two-dimensional materials that emit at telecom can also be efficiently addressed by NV centers. In my opinion, the experimental data are good, and the readout method and strategy are interesting for spin defects such as VB^- . The demonstration of nanoscale resolution of quantitatively mapping the spin defects beyond the optical diffraction limit is very useful. However, the idea of using NV to read out another electron spin has already been published in many works [Nat. Phys. 2013, 9, 215; Phys. Rev. Lett. 2014, 113 197601; Science 2015 347, 1135], and the novelty of using cross-relaxation is not significant enough. I think more data and discussion are needed to improve the manuscript before publication in Nature Communications.

1. The author mentioned that the cross relaxation method would "...eliminating the need for direct optical excitation, microwave driving, or emission-based readout" on page 3 and "this enables all-optical relaxometry-based ESR measurements" on page 14, which are exactly the advantages of cross relaxation and T1 relaxometry. However, the T1 relaxometry is also inefficient and not sensitive to a specific frequency of spin noise. Did the author consider the T2-based method, such as DEER and XY8 spectroscopy of NV? And in my opinion, it would be useful to add more discussion on comparing different NV-based strategies to prove the significance of cross-relaxation and T1 relaxometry in this work.

2. What is the dominant factor to determine the full width at half-maximum in NV single- τ T1-MR detection of VB^- , as shown in Fig. 2c and 2d? In Fig. 2d, there seems to be a double peak feature around -1.5 and 1.4; should it be a real spin structure or just a noisy feature limited by the current SNR? As mentioned by the author, the spin contrast of NV-based T1 relaxometry (such as Fig.2c, Fig.2d, and Fig.2f) is higher than that of VB^- 's ODMR (such as Fig.2a), but the SNR seems to be much lower, as marked by the large error bars in each data, which can be attributed to the low efficiency of T1 measurement. The author mentioned on page 13 that resolving this limitation by using an NV ensemble might decrease the spatial resolution of scanning NV. Can it be addressed by single-shot readout methods as shown in previous works [Nat. Commun. 2018, 9, 2406;]?

3. In Fig. 3b and 3f, the author maps the density of spin defects by Raman and scanning NV, respectively. Although it's obvious that the mapping region in 3f is much smaller than 3b. To make a clear comparison, it would be better to show a line profile in the figure. 3f and fit the step-like function.

4. In Figure 3a, the author mentioned that the hBN flakes were on the gold surface. Could the metal substrate bring extra noise to NV's T1 measurement? Meanwhile, the author applied Monte Carlo to show the relaxation rate of different NV-to-sample distances. It would be better to show a tip height-dependent T1 relaxation rate experimentally or comment more on this issue, since this would be important to justify the spatial resolution of T1-relaxometry of NV.

Also, on page 14, the author mentioned that "our technique primarily probes surface defect densities". Adding the data of tip height-dependent T1 relaxation rate would make this argument more compelling.

5. On page 11, the author cited 41 and 42 after "enabling direct imaging of magnetic noise and spin defect distributions with nanometric precision." However, the reference 42 mapped the static stray magnetic field of vortex states, instead of a

magnetic noise or spin defects. It would be better to change this reference here with a more suitable one.

Reviewer #2

(Remarks to the Author)
Please find my report attached.

Reviewer #3

(Remarks to the Author)
I co-reviewed this manuscript with one of the reviewers who provided the listed reports. This is part of the Nature Communications initiative to facilitate training in peer review and to provide appropriate recognition for Early Career Researchers who co-review manuscripts.

Reviewer #4

(Remarks to the Author)

Version 2:

Reviewer comments:

Reviewer #1

(Remarks to the Author)
The authors have thoroughly addressed all my comments raised in this round of review. They also provided additional data and explanations where required. In my opinion, the paper reads more clearly, and the analysis of the results is more convincing. I therefore recommend publication of "Probing Spin Defects via Single Spin Relaxometry" in Nature Communications.

Reviewer #2

(Remarks to the Author)

Reviewer #3

(Remarks to the Author)
I co-reviewed this manuscript with one of the reviewers who provided the listed reports. This is part of the Nature Communications initiative to facilitate training in peer review and to provide appropriate recognition for Early Career Researchers who co-review manuscripts.

Reviewer #4

(Remarks to the Author)
The authors have significantly revised the manuscript, including corrections to previous statements, the addition of supplementary data, and an improved summary. Overall, the paper is much improved compared to the previous version. In particular, the analysis of the tip-sample distance is now much more convincing, and I no longer have concerns regarding the importance or validity of the main findings.
The central result of this work is the demonstration of NV center relaxometry on ensembles of VB- spin defects, enabling an estimation of the near-surface spin density. Especially with the newly added gate-dependence data, the potential of T1 mapping of boron vacancy defects as a tool for studying hBN spin defects is now much clearer. The authors have addressed all my comments to my overall satisfaction. I recommend publication in Nat. Comms.
One minor comment concerns the title. The current title, "Probing spin defects...", is too broad and may be misleading to readers. I suggest that the authors revise the title to explicitly indicate that the work focuses on boron vacancy defects in hBN before publication.

REVIEWER COMMENTS

Reviewer #1

The manuscript titled "Probing Spin Defects via Single Spin Relaxometry" presents a novel study that utilizes scanning NV centers to read out the quantum states of VB^- and map its distribution. In this way, the low spin contrast of VB^- can be circumvented by relying on the high spin contrast and NV. Meanwhile, other spin-active quantum defects in two-dimensional materials that emit at telecom can also be efficiently addressed by NV centers. In my opinion, the experimental data are good, and the readout method and strategy are interesting for spin defects such as VB^- . The demonstration of nanoscale resolution of quantitatively mapping the spin defects beyond the optical diffraction limit is very useful.

Response 1.1: We thank the referee for the accurate summary of our work and the acknowledgement of the usefulness of our new method.

However, the idea of using NV to read out another electron spin has already been published in many works [Nat. Phys. 2013, 9, 215; Phys. Rev. Lett. 2014, 113 197601; Science 2015 347, 1135], and the novelty of using cross-relaxation is not significant enough. I think more data and discussion are needed to improve the manuscript before publication in Nature Communications.

Response 1.2: We thank the referee for raising the important question of novelty. We address this from two perspectives: (i) how our approach differs from previous NV–target-spin readout schemes, and (ii) why the cross-relaxation protocol has unique applications in the context of 2D spin defects, supported by new experiment added in Supplementary Note 8. We further emphasize that our method represents a fundamentally new defect-imaging capability compared to Raman or PL mapping.

Scanning NV cross-relaxometry enables non-invasive, charge-state-selective, quantitative imaging of VB^- defects with nanoscale spatial resolution, which is not accessible with existing optical techniques (see Response 4.7). This capability is timely and highly relevant given the rapid progress in 2D spin-based quantum sensors and the growing need for spatially resolved defect characterization in quantum materials.

First, we fully agree with the referee that using NV centers to read out other electron spins has been demonstrated in several pioneering works [Nat. Phys. **9**, 215 (2013); Phys. Rev. Lett. **113**, 197601 (2014); Science **347**, 1135 (2015)]. We have added these papers into the updated reference. However, these experiments rely on dynamical-decoupling protocols, which require (i) microwave driving of the NV center, (ii) microwave or RF driving of the target spin, (iii) prior knowledge of the target spin

resonance (frequency and π -pulse length), and (iv) nanosecond-scale timing control. In practice, such pulsed schemes are most convenient for frequencies up to the hundreds-of-MHz range, where high-power, short- π pulses are experimentally tractable.

In contrast, our method uses NV–VB⁻ cross-relaxation T₁ relaxometry as a **microwave-free, all-optical readout**. The protocol (i) does not require any microwave control of the NV center, (ii) only requires microsecond–millisecond timing resolution, (iii) operates with minimal prior knowledge of the target spin, and (iv) yields an ESR spectrum at GHz frequencies set by the applied magnetic field. Furthermore, by combining this cross-relaxation contrast with quantitative modeling, we can not only detect the presence of target spins but also map their local spin density at the nanoscale. To reflect the discussion, we have rewritten the following discussion section in main manuscript:

“Previous studies have demonstrated cross-relaxation (CR) between NV centers and nearby proton spins or other spin defects within diamond, and the ability of NV centers to read out other electron spins has been established in several pioneering works [Nat. Phys. 9, 215 (2013); Phys. Rev. Lett. 113, 197601 (2014); Science 347, 1135 (2015), Nat. Comm. 5, 4135 (2014), Phys. Rev. B 94 155402 (2016)]. However, these experiments rely on dynamical-decoupling protocols, which require (i) microwave driving of the NV center, (ii) microwave or RF driving of the target spin, (iii) prior knowledge of the target spin resonance (frequency and π -pulse length), and (iv) nanosecond-scale timing control; in practice, such pulsed schemes are most convenient for frequencies up to the hundreds-of-MHz range, where high-power, short π pulses are experimentally tractable. Cross-relaxation between an NV and spin-based quantum sensors in external materials—especially those of different dimensionalities—has not yet been achieved, and demonstrating such interactions is critical for extending NV-based detection to a broader range of quantum systems. Prior CR experiments have also relied on static diamond slabs, where the spatial relationship between the NV and target spins is fixed, limiting control over interaction strength and spatial resolution. In contrast, our method uses NV–VB⁻ cross-relaxation T₁ relaxometry as a microwave-free, all-optical readout: the protocol (i) does not require any microwave control of the NV center, (ii) only requires microsecond–millisecond timing resolution, (iii) operates with minimal prior knowledge of the target spin, and (iv) yields an ESR spectrum at GHz frequencies set by the applied magnetic field...”

Second, with regard to the novelty of the cross-relaxation application: to the best of our knowledge, **neither NV–spin relaxation involving a 2D spin-defect ensemble, nor its use for quantitative nanoscale defect-density mapping, has been demonstrated previously**. In our work, the external spins are VB⁻ defects in hBN, which are themselves promising spin-based quantum sensors. Our approach therefore realizes a **hybrid 3D–2D architecture** in which an NV center in diamond is used to

spectroscopically characterize and quantitatively map a 2D spin-defect sensor layer. This capability is particularly timely given the rapid development of 2D spin-active defects: other techniques either cannot distinguish VB^- from VB^0 or only access volume-averaged densities integrated over the full flake thickness, rather than the near-surface densities that are most relevant for device applications.

To further illustrate the unique information provided by our cross-relaxation method, **we have added a new experiment to the Supplementary Information (Note 8)**. A key open question in the hBN defect community is how effectively electrostatic gating can modulate the density of VB^- defects at the surface. Previous optical studies have reported conflicting results: one work [*Nano Lett.* **23**, 6141–6147 (2023)] finds essentially no gate dependence of VB^- density in relatively thick (~80 nm) flakes, while another reports only a few-percent change in VB^- PL intensity in thin hBN under gating [*Nano Lett.* **25**(14), 5836-5842 (2025)]. These PL-based measurements are inherently averaged over the full flake thickness, making it difficult to isolate a possible gate-induced modulation of the **surface** VB^- population.

Motivated by this, we hypothesize that gating primarily modifies the VB charge state near the surface through band bending, analogous to the $NV^- \leftrightarrow NV^0$ charge-state conversion in diamond [*Nat. Commun.* **3**, 729 (2012)]. Any gate-induced change, confined to a near-surface region, will be strongly diluted in PL measurements that integrate over the entire thickness, particularly in thicker flakes. Our scanning NV cross-relaxometry, by contrast, selectively probes the **surface-proximal VB^- spin density** that couples to the NV at the cross-relaxation field (see Figure S17). In the new data (SI Note 8), we apply a gate voltage while monitoring the NV T_1 relaxation rate at the NV– VB^- cross-relaxation condition, and we observe a gate-dependent change in the cross-relaxation-induced relaxation rate corresponding to an $\approx 30\%$ modulation of the near-surface VB^- density. This result is consistent with the surface-band-bending picture and demonstrates that our technique can **quantitatively and non-invasively access gate-tunable spin-defect densities in 2D materials**, which is difficult to achieve with existing approaches.

We have revised the Discussion to clearly articulate these methodological and application-level distinctions from previous NV–target-spin experiments and to highlight the new gate-dependent cross-relaxation data (Supplementary Note 8):

As a proof-of-principle relevant to device operation, we combine electrostatic gating with NV– VB^- cross-relaxometry (Supplementary Note 8) by tracking the gate-dependent change in NV T_1^{-1} at the cross-relaxation field, which directly reports the surface-proximal VB^- spin density. We observe an approximately 30% modulation, consistent with our surface band-bending hypothesis, in which electrostatic gating induces band bending that alters the local defect charge-state population in a shallow, surface-

proximal region of hBN (see Supplementary Note 8 for further discussion). This modulation is substantially larger than the few-percent changes inferred from thickness-averaged photoluminescence [Fraunie *et al.*, 2025], underscoring the charge-state selectivity and surface sensitivity of our non-invasive approach.

1. The author mentioned that the cross relaxation method would "...eliminating the need for direct optical excitation, microwave driving, or emission-based readout" on page 3 and "this enables all-optical relaxometry-based ESR measurements" on page 14, which are exactly the advantages of cross relaxation and T1 relaxometry. However, the T1 relaxometry is also inefficient and not sensitive to a specific frequency of spin noise. Did the author consider the T2-based method, such as DEER and XY8 spectroscopy of NV? And in my opinion, it would be useful to add more discussion on comparing different NV-based strategies to prove the significance of cross-relaxation and T1 relaxometry in this work.

Response 1.3: We agree that a comparison between T₁- and T₂-based NV sensing strategies is important. We have added a discussion in the main text discussion section that contrasts cross-relaxation T₁ relaxometry with dynamical-decoupling-based spectroscopy. As we explained in Response 1.2, in our experimental regime, the absence of microwave control, the prior knowledge about the target spin, and the nanosecond pulse requirements makes the T₂-based approach particularly convenient, especially at higher magnetic fields where NV ESR frequencies and the required π -pulse powers present practical challenges. We also point out that the cross-relaxation resonances remain frequency selective (bandwidth limited by T^{2*} of the target spin), so the method does not simply integrate over a broad noise spectrum.

2. What is the dominant factor to determine the full width at half-maximum in NV single-T1-MR detection of VB⁻, as shown in Fig. 2c and 2d? In Fig. 2d, there seems to be a double peak feature around -1.5 and 1.4; should it be a real spin structure or just a noisy feature limited by the current SNR? As mentioned by the author, the spin contrast of NV-based T1 relaxometry (such as Fig.2c, Fig.2d, and Fig.2f) is higher than that of VB⁻'s ODMR (such as Fig.2a), but the SNR seems to be much lower, as marked by the large error bars in each data, which can be attributed to the low efficiency of T1 measurement. The author mentioned on page 13 that resolving this limitation by using an NV ensemble might decrease the spatial resolution of scanning NV. Can it be addressed by single-shot readout methods as shown in previous works [Nat. Commun. 2018, 9, 2406;]?

Response 1.4. We thank the referee for this detailed question. In our regime, the T₁-MR linewidth is determined by the spectral overlap between the NV transition and the VB⁻ transition. This overlap can be viewed as the convolution of the NV and VB⁻ ESR

lineshapes, with an additional small contribution from the interaction-induced broadening γ (see Supplementary Note 4b). In practice, the dominant contribution to the full width at half-maximum (FWHM) of the single- τ T_1 -MR feature in Fig. 2c,d is the inhomogeneous linewidth of the VB^- ESR transition, which is set by the VB^- dephasing time T_2^* . This VB^- linewidth is significantly larger than both the intrinsic NV ESR linewidth (set by the NV T_2^*) and the interaction-induced broadening, so it controls the observed FWHM of the T_1 -MR dips. We have added a discussion on the T_1 -MR linewidth in updated Supplementary Note 4b.

The double-peak feature near ± 1.5 in Fig. 2d is not reproducible across repeated measurements and lies within the error bars of the data points. We therefore attribute it to residual noise rather than to a resolvable spin structure. To further support this interpretation, we have re-measured T_1 -MR on the same hBN_{nat} sample with a different NV probe and extended averaging; the resulting dataset (shown in new Supplementary Fig. S25) exhibits a single resonance without a double-peak feature, although it is still relatively noisy.

Regarding the signal-to-noise ratio, we agree that the error bars in Fig. 2c,d,f are larger than in the VB^- PL-ODMR spectra of Fig. 2a. This reflects the intrinsically lower efficiency of T_1 -based detection: each data point requires repeated initialize–wait–read cycles with wait times comparable to T_1 , which markedly reduces the number of photons collected per unit time compared to continuous-wave ODMR. On the other hand, after considering the fact that NV is significantly brighter than boron vacancies, NV T_1 -MR contrast is better than VB^- ODMR contrast, and the detector is much more efficient for NV wavelength (650-700 nm) compared to VB^- wavelength (850 nm), the difference between the efficiency of NV T_1 -MR technique and VB^- ODMR technique significantly narrows. We have added a detailed discussion on the T_1 -MR SNR in the new Supplementary Note 3 “Optimization and Sensitivity of T_1 -MR Measurements” considering all these factors. The discussion concludes with a note about the expected ability of the T_1 -MR method to excel in the few-spin limit when direct measurements of the VB^- PL are difficult or impractical. We have also added the following discussion into the main text:

“Supplementary Note 3 presents a quantitative model for single- τ T_1 -MR measurements and derives the optimal wait time τ that maximizes the signal-to-noise ratio by balancing contrast, photon shot noise, and measurement duty cycle. Within this framework, we evaluate the shot-noise-limited magnetic-field sensitivity near the NV– VB^- cross-relaxation condition. This analysis further indicates that T_1 -MR readout using the bright NV fluorescence can outperform direct PL-based ODMR of VB^- centers in the few-spin (or nanoscale-cluster) limit. In this regime, the VB^- photoluminescence from a ~ 1 –10 nm defect cluster can be far below the level typically detected in confocal

measurements that collect fluorescence from a diffraction-limited volume of ~500 nm lateral diameter. By contrast, scanning NV cross-relaxometry can place the NV directly above the cluster, and the cross-relaxation contribution to the NV relaxation rate is dominated by the nearest defects and decays steeply with separation ($\propto r^{-6}$), enabling a measurable T_1 -MR signal even when VB^- ODMR is not feasible.”

The single-shot / adaptive- τ readout schemes proposed in Ref. [Nat. Commun. **9**, 2406 (2018)] are indeed an attractive route to improving the efficiency of T_1 measurements. In that work, the largest speed-up occurs when the relaxation time varies over orders of magnitude and the protocol can dynamically adapt τ over a very wide range. In our experiments, the T_1 change at the cross-relaxation condition is more modest (usually it is approximately a factor of two), so the potential gain from a fully adaptive protocol is less dramatic. Moreover, implementing such adaptive samplings in a scanning NV experiment adds additional experimental complexity. We therefore chose a fixed- τ protocol for robustness and simplicity in the present work.

3. In Fig. 3b and 3f, the author maps the density of spin defects by Raman and scanning NV, respectively. Although it's obvious that the mapping region in 3f is much smaller than 3b. To make a clear comparison, it would be better to show a line profile in the figure. 3f and fit the step-like function.

Response 1.5. We thank the referee for this excellent suggestion. We have added an inset to Fig. 3f in which a line profile from the map is plotted and fit to error function, in which the extracted step width (46 nm) sheds light on the spatial resolution of this technique.

4. In Figure 3a, the author mentioned that the hBN flakes were on the gold surface. Could the metal substrate bring extra noise to NV's T_1 measurement? Meanwhile, the author applied Monte Carlo to show the relaxation rate of different NV-to-sample distances. It would be better to show a tip height-dependent T_1 relaxation rate experimentally or comment more on this issue, since this would be important to justify the spatial resolution of T_1 -relaxometry of NV.

Also, on page 14, the author mentioned that “our technique primarily probes surface defect densities”. Adding the data of tip height-dependent T_1 relaxation rate would make this argument more compelling.

Response 1.5. We thank the referee for pointing out the possible influence of the gold substrate on NV T_1 . Gold is used primarily in our initial VB^- ODMR characterization, where a coplanar waveguide (CPW) is required to deliver RF. In the scanning NV experiments, the hBN thickness (90–300 nm) strongly suppresses any magnetic noise from the metal.

Following the referee's suggestion, we have also performed T_1 measurements as a function of tip height above the hBN flake (new Figure S17, also copied here). The relaxation rate decreases monotonically with increasing height, consistent with a near-surface spin bath. These measurements support our statement that the technique primarily probes surface-proximal defect densities. We are also able to extract the initial NV-sample distance through fitting using the theory we developed in the updated Supplementary Note 4b. We have updated the main manuscript text and the SI accordingly.

Fig. S17: Height Dependence of NV T_1 at CR Condition: NV T_1 measured as a function of height above the hBN_{nat} surface at the CR condition ($B = 127$ G) as blue points. Fit to Eq. S52 as red curve.

5. On page 11, the author cited 41 and 42 after “enabling direct imaging of magnetic noise and spin defect distributions with nanometric precision.” However, the reference 42 mapped the static stray magnetic field of vortex states, instead of a magnetic noise or spin defects. It would be better to change this reference here with a more suitable one.

Response 1.6. We thank the referee for catching this. We have replaced Ref. 42 with [Sci. Adv. **5**, eaau8038 (2019)] that directly demonstrates imaging of magnetic noise / spin defects with shallowly implanted NV centers.

Reviewer #2

The manuscript by Melendez *et al.* presents experimental results of spin cross relaxation sensing of spin defects in hBN. Using shallow NV center in diamond tips as quantum sensors, they measure the spin resonance signal of VB⁻ centers, including the hyperfine structures from the host ¹⁴N and ¹⁵N nuclear spins. In addition, they demonstrate the image of spin defects in hBN flakes at the nanoscale.

Overall, this is an interesting and practical work. The advance of the study can be identified as the experimental characterization of spin defects on the surface of 2D materials.

Response 2.1. We thank the referee for the accurate summary of our work.

However, the manuscript also raises some critical issues, I therefore suggest a major revision for further consideration.

Below are more specific comments:

1. The method presented in this paper is very similar to scanning NV relaxometry, which is used to study magnetic noise. For example, Finco *et al.* [Nat. Commun. 12, 767 (2021)] have used scanning NV single-spin relaxometry to image non-collinear antiferromagnetic textures, and Brecht *et al.* [Nano Lett. 21, 8213 (2021)] have used relaxation rates to study magnon densities. We therefore suggest the authors to explain the novelty of the current work in comparison to the previous studies.

Response 2.2. We appreciate the referee's suggestion to better position our work relative to prior scanning NV relaxometry experiments. Finco *et al.* [Nat. Commun. 12, 767 (2021)] and Brecht *et al.* [Nano Lett. 21, 8213 (2021)] used single-spin relaxometry to image non-collinear antiferromagnetic textures and magnon densities, respectively, in *magnetic* materials. In those works, the NV center probes magnetic noise associated with an underlying magnetic order parameter or magnon bath.

In contrast, our experiments target a **two-dimensional ensemble of VB⁻ defects in hBN**, which serve as a prototype **non-magnetic** spin-based **quantum sensor** rather than a magnetic order parameter. By operating at the **NV–VB⁻ cross-relaxation condition**, we (i) **obtain a local ESR spectrum of VB⁻ with resolvable hyperfine structure** and (ii) use the cross-relaxation contrast to **quantitatively extract VB⁻ spin densities** and map them at the nanoscale. Thus, compared to Refs. [Finco 2021; Brecht 2021], which use scanning NV relaxometry to image magnetic textures and magnon populations, our work demonstrates **scanning NV cross-relaxometry of a 2D**

spin-defect ensemble, providing both spatially resolved **ESR spectroscopy and quantitative spin-density mapping** of an external quantum-sensor material. In contrast, the previous relaxometry map cannot offer simultaneously offer information of spectrum and quantitative spin density.

A more detailed discussion of the novelty and an example of a unique application of our technique (probing gate-dependent modulation of near-surface VB^- density) is provided in our response to Referee 1, Comment 1 (Response 1.2), and in the revised Discussion section of the manuscript.

Another novelty is that our approach provides a fundamentally new defect-imaging capability beyond Raman or PL mapping, enabling **non-invasive, charge-state-selective, quantitative imaging of VB^- defects with nanoscale spatial resolution** (see Response 4.7). This capability is timely and significant in light of the rapid development of 2D spin-based quantum sensors and the pressing need for spatially resolved defect characterization in quantum materials.

2. I am a little confused about the introduction of the quantum network. The current experimental results have little to do with the applications of quantum networks. On the one hand, the spin coherence and optical properties of the VB centers in hBN are far from the requirements for a quantum network. On the other hand, the observed cross-relaxation signal only means that the two spin systems are coupled, which is far from transferring a quantum state or quantum information between them.

Response 2.3. We agree that our current experimental results are not yet at the stage of demonstrating quantum networking. **In the revised introduction section**, we have removed the quantum-network terminology and instead frame the work in terms of heterogeneous quantum sensing: using an NV sensor to characterize and benchmark spin defects in 2D materials that may be useful for future quantum technologies.

3. NV centers with different spin relaxation time (T_1) were used for the experiment. What is the strategy for selecting NV centers to perform the cross-relaxation measurement? Meanwhile, why is the measured T_1 , even before exposure to the sample, significantly shorter than the typical T_1 of NV centers in bulk diamond?

Response 2.4. We thank the referee for this question. Our cross-relaxation measurements were performed with several scanning NV probes. We first screen candidate NV centers on each probe chip by confocal characterization, and select those that simultaneously provide (i) stable NV- charge state and photostable PL, (ii) clear ODMR contrast and well-resolved NV resonances, and (iii) relatively short T_1 to

enable practical acquisition times for multi-point T_1 -MR spectra and maps. In addition, we select NV tips with flat tip surface. For the 6 keV implanted NV centers used in this work, the measured T_1 values typically fall in the ~ 2 – 8 ms range. For cross-relaxation experiment, we select the NV centers with shorter T_1 (usually below 5 ms, see new Supplementary Note 1 for more discussion including the strategy of selecting an NV tip, and Supplementary Table S1 for a summary of the tips used in this study).

Regarding the absolute T_1 : we agree it is shorter than the T_1 of NV centers in bulk diamond. This is expected because the scanning probes employ shallow implanted NV centers close to the diamond surface ($\sim 9 \pm 4$ nm underneath the surface), where additional relaxation channels—e.g., magnetic noise from surface paramagnetic states, implantation-induced defects, and surface-related charge/spin fluctuations—substantially enhance the relaxation rate compared to bulk. Therefore, the “short raw T_1 ” primarily reflects the shallow-depth, nanofabricated scanning-probe environment rather than any sample-induced effect. The second reason is that we purposely chose shorter T_1 NV centers to reduce data acquisition time.

Finally, NV–sample distance is a critical parameter for cross-relaxation strength. In practice, we favor probes that minimize the effective NV–sample separation, which depends on both (a) the NV depth below the diamond surface and (b) the mechanical standoff set by probe geometry. To reduce standoff, we use truncated-paraboloid probe geometries that provide a flatter apex region and more reliable close approach than conventional tapered tips (see updated Supplementary Note 1).

4. In the T_1 -MR spectrum shown in Fig. 2f, the four resonance dips appear to have equal amplitudes, which is in contrast to the ratio of 1:3:3:1 observed in the ODMR spectrum. Is this consistent with theoretical expectations?

Response 2.5. We thank the reviewer for raising this point. We agree that, in an idealized weak-coupling picture with fully resolved hyperfine lines and purely linear response, one might expect the T_1 -MR dip amplitudes to reflect the same relative transition strengths that give rise to the 1:3:3:1 ODMR ratio. In our experiment, however, the T_1 -MR data in Fig. 2f were acquired using a short- T_1 NV tip (~ 191 μ s), which is likely to also have a short T_2 and a noisy surface. Under these conditions the relative dip depths can be distorted by several non-ideal factors, including broadened NV spectral response, overlap between neighboring hyperfine features, and saturation/nonlinearities in the cross-relaxation process. These effects can wash out the expected amplitude ratios and lead to more uniform apparent dip depths.

We emphasize that our key conclusions do not rely on the relative amplitudes of the dips: the T_1 -MR spectrum robustly resolves the **resonance frequencies** and **linewidths** of the hyperfine-split transitions, which are the quantities used in our analysis and are not affected by an overall rescaling or partial saturation of the dip contrast. We therefore interpret the near-equal dip depths as an experimental response-function effect rather than a discrepancy in the extracted ESR frequencies or linewidths, and we have revised the text to clarify this point:

“We note that the relative dip depths in T_1 -MR can deviate from the hyperfine intensity ratios observed in direct ODMR, particularly for short- T_1 probes where a broadened response and partial saturation/nonlinear cross-relaxation can equalize the apparent contrasts. Importantly, these effects do not affect the extracted resonance frequencies or linewidths, which underpin the analysis.”

5. It would be better if you provided more details on the experimental methods and data processing, preferably in the method or SI, so that others can easily understand and reproduce the results. Critical details include the diamond tip (e.g. NV count rate, magnetic sensitivity), choice of τ in the single- τ measurements, calibration of the external magnetic field, etc.

Response 2.6. We thank the referee for pointing out the need for clarity on these subjects.

The typical diamond tip parameters are included in updated Supplementary Note 1 and the choice of τ is discussed in new Supplementary Note 3. In this discussion we have calculated the SNR and magnetic field sensitivity of the single τ T_1 -MR measurement as a function of τ allowing for optimization with respect to the wait time τ . The calibration of external field is relatively straightforward, which is based on ODMR of NV and/or boron vacancies.

Other minor issues

- In Fig. 1d, the horizontal axis of the T_1 measurement stops before the signal decays completely. Please indicate the fitting function used and the stretch exponent (if applicable).

Response 2.6. We thank the referee for pointing out the need for emphasis of this point. At the end of the second paragraph in the Sensing Protocols subsection within the Materials and Methods section, we have modified the wording to better emphasize that fitting was done to an exponential $\exp(-t/T_1)$ without stretching. In addition, a sentence has been added to the caption of Fig. 1d explicitly stating the same point. We also

added more T1 data before and after measured at cross relaxation conditions to confirm the cross-relaxation effect (Table S1).

- The authors write “Figure 2 were linearly baseline-corrected to isolate the contrast associated with cross-relaxation.” Please describe how this was done.

Response 2.7. We thank the referee for requesting clarification. The linear baseline correction was introduced to remove a smooth, non-resonant background arising from the intrinsic field dependence of the NV readout contrast in the single- τ T_1 -MR measurement. Specifically, over the magnetic-field range used in Fig. 2, the ODMR/ T_1 readout contrast of the NV $m_s = 0 \leftrightarrow +1$ transition decreases approximately linearly with increasing field (see Fig. S21). This field-dependent reduction in contrast produces a correspondingly smooth baseline variation in the single- τ T_1 -MR signal that is unrelated to cross-relaxation.

To account for this effect, we use the measured (approximately linear) contrast–field dependence to derive the expected baseline form of the single- τ T_1 -MR signal. As shown in the new Supplementary Note 5 (Eqs. S61–S64), if the readout contrast depends linearly on magnetic field, then the resulting single- τ T_1 -MR signal also acquires an approximately linear field dependence. Therefore, for each single- τ T_1 -MR trace in Fig. 2, we fit the non-resonant background with a linear function and subtract this fitted baseline. This procedure isolates field-dependent T1 contrast features associated with NV-VB⁻ cross-relaxation, without altering their spectral positions or widths. We now describe this baseline-correction procedure explicitly in the revised main text and refer readers to Supplementary Note~5 for the derivation. The updated main text is:

“However, the $PL(\tau)/PL(0)$ signal is also influenced by the NV readout contrast, which in our experiments decreases approximately linearly with increasing magnetic field (Supplementary Fig. S21). This introduces a smooth, non-resonant background in the single- τ T_1 -MR traces that is unrelated to cross-relaxation. To remove this contribution, each T_1 -MR curve in Fig. 2 was baseline-corrected by fitting the non-resonant background to a linear function and subtracting it, thereby isolating the cross-relaxation features. A full derivation and detailed description of this baseline-correction procedure are provided in Supplementary Note 5.”

- Regarding the statement of “In addition, unlike optical methods that integrate signals over the entire sample thickness, our technique primarily probes surface defect densities...”, please provide a quantitative estimate of the depth of the sensing volume.

Response 2.8. We thank the referee for this helpful suggestion. To quantitatively estimate the sensing volume, we performed a finite-size analysis of the Monte Carlo simulations at the extracted density of ~220 ppm (see new Supplementary Note 4 and Fig. S16). We find that the NV–VB⁻ cross-relaxation rate saturates once the simulated bath size exceeds ~5000 VB⁻ spins. This corresponds to spins extending up to ~50 nm away from the NV sensor, confined within a spherical cap geometry defined by the sensing sphere intersecting the hBN layer. The resulting effective sensing volume is ~1.8 × 10⁵ nm³. This refined analysis improves upon our earlier simulations (1000 spins) and provides a more accurate density estimate, while remaining consistent with our experimental observations. We thank the referee for pointing out this aspect, which has helped us refine our analysis. In addition, in our new Figure S17, we have experimentally measured how quickly the cross-relaxation signal decays with NV tip lift height, further proving that our technique mainly probes the surface. We now include these updated simulations and discussion in the revision.

- The Monte Carlo simulations considered a fixed number of VB⁻ spins at different densities. Could the authors clarify how the spatial scale (or cutoff) of the interacting spins was chosen and whether this choice affects the simulation results?

Response 2.9. We also thank the referee for raising this important point. The finite-size analysis described above directly addresses the question of the spatial cutoff in our simulations. We find that the cross-relaxation rate converges once ~5000 spins (up to ~50 nm from the NV) are included, establishing the appropriate cutoff scale. Beyond this range, extending the bath size does not significantly alter the results, confirming that the extracted densities are robust. The discussion is presented in new Supplementary Note 4b.

Reviewer #3

We thank Reviewer #3 for their helpful comments and contributions as a co-reviewer

Reviewer #4

The manuscript titled “Probing Spin Defects via Single Spin Relaxometry” by Melendez et al. applies scanning NV magnetometry to study spin defects in naturally and isotopically purified hBN. The authors observed a decrease in the T1 time of the single NV center in the diamond tip when brought close to VB⁻ spin ensembles under an external field at which the spin state transitions of NV and VB⁻ match. The increase of the relaxation rate when approximate to a spin ensemble is expectable; however, the

impressive aspects one is that hyperfine coupling could also be observed due to the small NV–sample distance. The second one is the T1 relaxation map of natural hBN was also able to image and quantitatively resolve VB^- density at the nanoscale, which is crucial for optimizing spin defects in 2D materials for quantum sensing. The observation is noteworthy.

Response 4.1. We thank the referee for the accurate summary of our work and the acknowledgement of the impressive aspects of our new method.

however, there remain some ambiguities in the presentation of the data and methodology in this form of the manuscript. Also, I remain uncertain about the potential significance of the results. Therefore, I cannot recommend it for publication in its current form in Nature Communications.

Response 4.2. A more detailed discussion of the novelty and an example of a unique application of our technique (probing gate-dependent modulation of near-surface VB^- density) is provided in our response to Referee 1, Comment 1 (Response 1.2), and in the revised Discussion section of the manuscript. In response 4.7 (response to the 5th comments here), we also explain the advantages of our technique compared to other defect mapping methods (e.g., Raman). We therefore hope the referee finds our technique useful.

Major concerns are listed below:

1. The authors state that the NV–sample distance is 11.4 ± 1.5 nm, which is one of the closest NV to sample distances reported. However, this analysis was only based on a rough $B \sim 1/d$ scaling from two linecuts in lift-up mode. The magnetization and NV to sample distance could be directly obtained from linecut fitting (Thiel Science (2019) or ref. 3 in the SI), instead of using the linewidth and amplitude separately. I understand that extracting the NV to sample distance is challenging, but I was not convinced by the current analysis. Moreover, the comprehensive analysis of spin density depends on this distance, especially given that an increased NV to sample distance would lead to dramatic changes in both the central value and the range of the estimated spin density

Response 4.3. We thank the referee for this important point. We agree that the NV–sample distance is a key parameter for quantitative spin-density extraction, and we

have revised Supplementary Note 1 to clarify how this distance was determined and validated.

First, the quoted NV–sample surface distance of 11.4 ± 1.5 nm is not based solely on a rough $B \propto 1/d$ estimate. In the revised SI note 1 we explicitly describe that probe #1 was calibrated on a well-defined Ta/CoFeB/Ta stripe structure using lift-mode ODMR line scans at two lift heights (80 nm and 120 nm). From the extracted linecut(s), we use the established stripe-field model (SI Refs. 2) and extract the NV–sample distance via amplitude scaling ($B \sim 1/d$), obtaining 13.9 ± 1.5 nm for the NV-to-capping-layer distance and 11.4 ± 1.5 nm after subtracting the 2.5 nm capping layer. Second, **as an independent cross-check**, we also fit the magnetic feature linewidth in the 80 nm lift-height line scan using the same analytic model (SI Ref. 2, which is the “linecut fitting” mentioned by the referee), which yields an average NV–sample distance of 12.9 ± 1.9 nm. Since 11.4 ± 1.5 nm and 12.9 ± 1.9 nm are similar, we tend to believe these numbers. In addition, we note in the SI that this linewidth-based method is sensitive to probe geometry as well as the surface smoothness of the magnet, and can overestimate the separation, so we treat it as a consistency check rather than our primary estimate.

Third, we expanded Supplementary Note 1 to explain the experimental steps taken to reliably reach small standoff distances, including (i) screening >10 probes and selecting tips with short T_1 and flat end facets, (ii) PL-assisted contact optimization using V_B^- collection through the nanopillar, (iii) reverse imaging and cleaning using a TGT1 grating to verify and maintain a flat tip surface, and (iv) using a special flat tip.

Finally, to address the referee’s concern that the extracted spin density depends on the assumed standoff distance, we now (i) clearly identify which probe and which calibration method corresponds to each dataset via Table S1, and (ii) provide additional distance determinations on other probes (including probe #4) to demonstrate that our overall conclusions do not rely on a single-tip assumption. The NV-distance of probe #4 was determined using two different methods (scanning a magnet stripe & using height-dependent relaxometry) but resulted in consistent values, showing our method works correctly.

We rewrote SI Note 1 to reflect all these points. Specifically, we have this writing to clarify that we did the line scan fitting as well:

“As an independent estimate of the NV–sample distance, we analyzed the linewidths of magnetic features measured in the 80 nm lift-height scan, fitting the data using the model described in Ref. [Phys. Rev. Applied 4, 014003]. From the left magnetic peak, we obtained an apparent distance of 94.5 ± 0.5 nm, corresponding to 12.0 ± 0.5 nm after accounting for the 80 nm lift height and the capping layer. Fitting the right peak yielded a distance of 13.8 ± 2.6 nm, resulting in an average NV–sample separation of

12.9 ± 1.9 nm. Because this linewidth-based approach is sensitive to probe geometry and tends to overestimate the true separation (Ref. [ACS Nano, doi:10.1021/acsnano.4c18460]), we do not rely on it as our primary distance estimate.”

2. A related question, what parameters were used for the NV field in the simulations shown in Fig. S2 (e.g., NV to sample distance, magnetization density)? Why is the 80 nm

curve nearly twice the value presented in Fig. S1?

Response 4.4. We thank the referee for pointing this out. We agree that the original simulation panel could be confusing because it was intended only to illustrate qualitative scaling and did not use the exact experimental parameters (e.g., magnetization and NV–sample distance). In the revised SI, we removed this simulation and instead cite prior literature that reports the same analytic field scaling and linecut-fitting approach used here (SI Refs. 1–2). This eliminates ambiguity and keeps the SI fully quantitative and self-consistent.

3. Since the NV to sample distance normally varies significantly from tip to tip (ref. 3 in the

SI), how many tips were used in this study? Can I understand that all the data, except for

Fig. 2e–f, were obtained with the same tip, for which the NV height was analyzed? The NV implantation depth is 9 ± 4 nm. Is this range derived from SRIM simulations or from examining different tips?

Response 4.5. We thank the referee for raising these points. In this study, the cross-relaxometry and control datasets were acquired using **four** different NV probes. We have added Table S1 to the SI, which explicitly maps each figure panel to the corresponding probe number and summarizes key probe properties (including T_1 and, where measured, the NV–sample distance).

Regarding the implantation depth, the value 9 ± 4 nm corresponds to the expected mean depth for 6 keV nitrogen implantation based on SRIM simulations, and we now state this explicitly in Supplementary Note 1.

4. Could the authors provide a comment on the spin density of the $h^{10}B^{15}N$ sample compared to the natural hBN? What is the thickness of the flake? I also did not see a strong logical connection between Fig. 2 and the other figures.

Response 4.6. We thank the referee for these questions and have revised the SI and Methods to clarify (i) the defect/spin density comparison between isotopically purified $h^{10}\text{B}^{15}\text{N}$ and natural-abundance hBN, and (ii) the flake thickness used in each dataset.

Spin density comparison. For natural-abundance hBN irradiated at a high dose ($50 \text{ He}^+/\text{nm}^2$), literature reports a total boron-vacancy density (including both VB^- and VB^0) on the order of $\sim 5400\text{ppm}$. In contrast, our NV relaxometry and cross-relaxometry measurements selectively probe the paramagnetic VB^- population. Under comparable conditions, we extract a VB^- spin density of $\sim 200 - 400 \text{ ppm}$, indicating that only a minority of boron vacancies occupy the VB^- charge state.

For isotopically purified $h^{10}\text{B}^{15}\text{N}$, the VB^- density is primarily set by the neutron irradiation dose (approximately linear with fluence before saturation). For the sample used here (neutron fluence $1.4 \times 10^{16} \text{ cm}^{-2}$), we determine a VB^- spin density of $\sim 40 \text{ ppm}$ (new Fig. S26).

Flake thickness. The exfoliated $h^{10}\text{B}^{15}\text{N}$ flakes used in this work are typically 10–50 nm thick. We now state the thickness used in the relevant datasets explicitly: the measurements in Fig. 2 were performed on a $\sim 50\text{nm}$ flake, and the data in Fig. S26 were obtained on a $\sim 15\text{nm}$ flake (see the revised Methods/ Supplementary Note 8). For example, in the update method section, we have:

“Prior to the cross-relaxometry experiments, micrometer-scale flakes were exfoliated from irradiated macroscopic crystals using standard mechanical exfoliation. This process yielded flakes with thicknesses of approximately 50 nm, used for the measurements shown in Fig. 2, and approximately 15 nm, used for the data presented in Fig. S26.”

Logical connection between Fig. 2 and the rest of the manuscript. The central goal of the manuscript is to demonstrate scanning NV cross-relaxometry as a general method to selectively detect, spectroscopically identify, and spatially map proximal spin defects. The narrative is structured as follows:

- (i) Fig. 1 establishes the existence and selectivity of NV– VB^- cross-relaxation.
- (ii) Fig. 2 demonstrates that cross-relaxometry provides local ESR spectroscopy, resolving spectral features (including hyperfine structure) at a single location in isotopically purified material.
- (iii) Fig. 3 builds directly on this spectroscopic capability by combining modeling with spatially resolved T_1 maps to quantitatively image the nanoscale VB^- spin-density distribution.

Thus, Fig. 2 provides the spectroscopic foundation (local identification and spectral fingerprints) that motivates and supports the quantitative mapping demonstrated in Fig.

3. In addition, SI Note 8 further extends the method to a gate-dependent study, illustrating its applicability to probing surface charge-state conversion of VB defects.

5. Another main question is what additional information the relaxation map in Fig. 3f provides. The map only shows two steps of spin density in regions with large height differences, but does not show any gradual change for different thicknesses or variations

in defect density (which can be seen in the Raman map). An important point is whether the relaxation maps provide additional information compared to the Raman map (e.g., in the upper-left corner of Fig. 3b, where the Raman data show a discontinuity within a region of uniform thickness) or compared to the counts in confocal PL map (Fig. S5).

Response 4.7. We thank the referee for this important question and have revised the manuscript and SI to clarify what information Fig. 3f provides beyond Raman and confocal PL.

First, the apparent “two-step” behavior in Fig. 3f is consistent with sharp spatial transitions in VB⁻ density across grain boundaries rather than a gradual density variation. In CVD-grown samples, such boundaries can be atomically abrupt. To substantiate this point, we added a high-resolution AFM image (new Fig. S27b) showing that the relevant boundary is sharp. We also added a linecut analysis of the NV-derived density map as an inset to Fig. 3f, yielding an edge width of 46 nm from fitting. By contrast, Raman and confocal PL maps (new Figure S27a) appear more gradual primarily due to their diffraction-limited spatial resolution (typically ~500 nm) and the fact that they are averaging signal across the entire ~90-240 nm sample thickness instead of probing the surface only, which spatially average an intrinsically sharp transition.

Second, regarding whether the relaxation map provides additional information compared to Raman, we emphasize that cross-relaxometry offers four capabilities that Raman map does not provide simultaneously:

- (i) charge-state selectivity (VB⁻ only), while Raman cannot distinguish between VB⁻ and VB⁰
- (ii) sub-diffraction spatial resolution,
- (iii) surface sensitivity (see new Figure S17), and
- (iv) quantitative defect density.

This new imaging modality is timely and significant in light of the rapid development of 2D spin-based quantum sensors and the pressing need for spatially resolved defect characterization in quantum materials.

Fig. S27: Gate-Dependence of V_B^- Surface Density: Additional imaging of the CVD-grown hBN_{nat} sample shown in Fig. 3. (a) Confocal PL intensity map of the sample. (b) High-resolution AFM topography of the top-right region corresponding to the area imaged in Fig. 3f.

6. Is the gold substrate used to create a larger spin density contrast with respect to sample

thickness? If so, how would this differ compared to a standard substrate such as SiO_2 ?

Response 4.8. For CVD hBN_{nat} sample, gold is used primarily in our initial V_B^- ODMR characterization, where a gold/sapphire coplanar waveguide (CPW) is required to deliver RF. So having a gold substrate rather than SiO_2 is desired in this experiment.

In terms of creating a large spin density contrast with respect to sample thickness during He ion irradiation, from our experiment and simulation, yes, the choice of an Au underlayer can enhance thickness-dependent defect creation under He^+ irradiation, primarily because **Au is a heavy element and a high-Z material** with a much larger elastic backscattering probability for He ions than Si or SiO_2 . In Rutherford-type backscattering, the scattering cross-section scales approximately as Z^2 , so heavy elements such as Au generate substantially higher backscatter yields than light substrates (e.g., Si, O). See [*Journal of Applied Physics* 109.6 (2011)] for more information.

In our SRIM-based picture (Fig. S22), thin hBN (e.g., ~90 nm) allows a significant fraction of incident He ions to reach the Au film; a portion of these ions are then backscattered into the hBN, increasing the effective ion path length and modifying where energy is deposited and where vacancies are generated near the near-surface region. By contrast, for a standard SiO₂/Si substrate, the backscattered He yield is expected to be significantly lower because the substrate is much lower *Z* than Au; consequently, fewer ions would be reflected back into the hBN, and the thickness-dependent “substrate backscatter” component would be weaker. This expectation is consistent with the broader helium-ion microscopy literature noting that heavy elements scatter helium at much higher rates than silicon [*J. Vac. Sci. Technol. B* 26, 2103–2106 (2008)].

We have revised Supplementary Note 6 to clarify that Au can enhance the contrast of boron vacancy density when using He ion irradiation especially when the hBN thickness is shorter than the He ion penetration depth.

Some minor comments:

a. The arrow in Fig. 1c is not described in the caption.

A sentence has been added in the caption of Fig. 1c describing the arrows. We appreciate this correction.

b. At what field was the data for the non CR T1 curve taken? Is there any influence due to the field not being aligned with the NV center?

The field was 32 G. During our experiment, when field is not aligned with NV axis, the transverse field is within 100 G. At this relatively small field we did not observe a measurable NV T1 change. From literature, if the field is sufficiently large (beyond a few hundred Gauss), the relaxation rate shows approximately linear dependence on the field [*Nat Commun* 10, 5160 (2019)]. Although we are not using a large field, at larger field, as long as the T1 contrast roughly decreases linearly with increased field, we can still perform T1-MR experiment after the linear background correction of the T1-MR data (see new Supplementary Note 5, Eqs. S61—S64). On the other hand, the transverse field could reduce NV ODMR or T1 contrast (see Figure S21).

c. In the main text, the average hyperfine splitting is reported as $A_{zz} = 2\pi \times (67 \pm 2)$ MHz, whereas in the caption of Fig. 2 it is given as $A_{zz} = 63$ MHz. And $A_{zz} = 65$ MHz in Fig. S17.

We thank the referee for pointing out an apparent discrepancy in our reporting of the h¹⁰B¹⁵N hyperfine splitting parameter A_{zz} . The $A_{zz}=67$ MHz reported in the main text on Page 8 is the average A_{zz} for the h¹⁰B¹⁵N sample that was measured in Fig. 2. The

value of 65 MHz was measured on a different sample from the same batch which corresponds to the data in Fig. S24. The discrepancy is not dissimilar to the range of values reported in literature, such as 65.9 ± 0.9 MHz [Gong, R., Nat Commun 15, 104 (2024)] and 64.1(2) [Clua-Provost, T., Phys. Rev. Lett. 131, 126901 (2023)]. Lastly, the value of 63 MHz came from a preliminary measurement which we ended up not including and had accidentally forgotten to remove. The correct value should have been 67 MHz and has since been corrected. We greatly appreciate the referee pointing out the typo. **We have updated the main text to correct this mistake.**

Reviewer #1:

The authors have thoroughly addressed all my comments raised in this round of review. They also provided additional data and explanations where required. In my opinion, the paper reads more clearly, and the analysis of the results is more convincing. I therefore recommend publication of “Probing Spin Defects via Single Spin Relaxometry” in Nature Communications.

We thank the referee for their guidance in improving our manuscript.

Reviewer #2:

The revised manuscript clarifies the novelty of the work, specifically imaging the density of surface spin defects by combining the CR method with a scanning NV probe. Most of the issues raised by the reviewers have been properly addressed. I suggest a minor revision to address the following points.

1. In the second paragraph of the Introduction, the authors write, “a major limitation remains: spin-active quantum defects in two-dimensional materials that emit at telecom wavelengths...” As the CR method is unrelated to the optical properties of the target spin, I suggest removing or revising this point. Instead, the rationale for the need for nanoscale spin density imaging should be included.

We thank the referee for this valid point. We have revised the second half of the second paragraph of the introduction as follows:

“Currently, hexagonal boron nitride (hBN) is the only known two-dimensional material hosting optically active spin defects, making it uniquely promising for surface-proximal quantum sensing (24, 25). In addition to hBN with naturally occurring isotope ratios (hBN_{nat}), isotopically purified h¹⁰B¹⁵N allows clear observation of the hyperfine structure, giving insight into electron-nuclear interactions as well as an increase in magnetic field sensitivity (26, 27). More broadly, the discovery and characterization of new spin-active defects in low-dimensional materials—some of which may ultimately emit in technologically relevant spectral ranges such as the near-infrared or telecom—requires experimental tools that do not rely on defect-specific optical readout. A key outstanding challenge is therefore the ability to directly image the spatial distribution, density, and charge state of spin-active defects at the nanoscale. Conventional optical and structural probes are typically diffraction-limited, thickness-averaged, or unable to distinguish spin-active from spin-inactive charge states, masking nanoscale inhomogeneities that govern

decoherence, spin–spin interactions, and device functionality. These limitations motivate the development of a nanoscale, charge-state-selective spin-density imaging technique that is independent of the optical properties of the target spin.”

With this revision we have shifted the focus on the need for nanoscale, charge-state-selective spin density imaging as the primary motivation, and reframed the optical properties of the spin as a possible downstream benefit of CR-based detection.

2. Regarding the measurement of spin defect density, please provide a quantitative estimate of the performance of the CR method. For example, is it possible to detect a single V_B^- center in hBN? If not, what is the minimum detectable density? It appears that the intrinsic relaxation rate of the NV spin and the distance between the NV probe and the sample spin are important factors.

We thank the referee for pointing out the need for a quantitative discussion on this matter. We have added to the end of Supplemental Note 3 with a discussion about the minimum detectable density, as well as the prospect of single-spin detection given the NV-to-sample distance:

“To quantify the performance limits of the cross-relaxation (CR) method, we provide a quantitative estimate of the minimum detectable defect density and the feasibility of single-spin detection based on the experimental parameters derived in Supplementary Note 4. First, regarding the minimum detectable volume density, we consider the SNR in the measurement of the induced relaxation rate change. In our experiments, a V_B^- density of approximately 220 ppm yielded an induced relaxation rate change of $\Gamma_1^{CR} \approx 1.72$ kHz at an NV-sample distance of 11.4 nm. Under typical experimental conditions (integration time $T_m \approx 60$ s, contrast $C \approx 0.15$), the minimum resolvable change in the relaxation rate is determined by photon shot noise to be $\delta\Gamma \approx 10$ -20 Hz. Assuming a linear scaling of the induced Γ_1^{CR} with density in the dilute limit (Eq.~(S56)), this noise floor implies a minimum detectable V_B^- concentration of approximately 1-3 ppm.

Second, we evaluate the feasibility of detecting a single V_B^- center by considering the coupling rate $\Gamma_{1,j}$ between the NV and a single defect at distance r . Following the derivation in Supplementary Note 4, the rate scales as $\Gamma_{1,j} \propto 1 / (r^6 \times \Gamma_{tot})$, where Γ_{tot} is the effective spin linewidth. Assuming a V_B^- linewidth of $\Gamma_{tot} \approx 2\pi \times 200$ MHz (consistent with the broad ODMR features in Fig. S8), a single V_B^- defect at $r = 10$ nm is estimated to induce a relaxation rate change of $\Gamma_{1,j} \approx 10$ -15 Hz. For a high-quality probe

with an intrinsic relaxation rate $\Gamma_1^{\text{int}} \approx 200$ Hz, this represents a ~5-8% change relative to the background rate. If we consider the narrower hyperfine features (~10s of MHz) observed in Fig. S18, the coupling increases to $\Gamma_{1,j} \approx 20$ -50 Hz, corresponding to up to a ~10-25% change in the ideal case.

While a broader linewidth reduces the peak signal, the $1/r^6$ dependence ensures the signal increases dramatically at shorter distances. For example, at an NV-sample distance of $d=8$ nm with a 200 MHz linewidth, the single-spin signal rises to ~40-60 Hz, making it resolvable even against a higher noise floor ($\Gamma_1^{\text{int}} \sim 1$ kHz). Notably, the estimated minimum detectable density (1-3 ppm) is physically consistent with the single-spin limit. In hBN, a 1 ppm concentration corresponds to approximately 0.1 spins/nm³. Given the $1/r^6$ locality of the dipolar interaction, the NV sensor effectively probes a volume of $V_{\text{eff}} \approx (2/3)\pi d^3$. For $d \approx 11.4$ nm this volume contains on average less than one V_B^- defect at 1 ppm. Thus, the sensitivity threshold of the ensemble measurement naturally coincides with the regime where the signal is dominated by the single nearest defect spin. We conclude that single-spin resolution is achievable provided the standoff distance is maintained at or below ~10 nm.”

With this addition we believe the manuscript to be further strengthened in terms of CR-based spin sensing, as well as offering possible directions for future research.

3. You mentioned the NV center (V_B^- center) as 3D (2D) spin sensors; in fact, they are both 0D sensors. A more accurate description is spin sensors in 3D or 2D materials.

We thank the referee for suggesting this clarification. We agree that this is an important distinction that should be emphasized. We have modified the last sentence in our abstract to now read:

“Our method demonstrates interactions between spin sensors in 3D and 2D materials, establishing NV centers as versatile probes for characterizing otherwise inaccessible spin defects.”

4. In Abstract, the authors write “...with scanning probe microscopy to discover, read out, and spatially map arbitrary spin-based quantum sensors at the nanoscale.” Please remove the word “arbitrary”, as only resonant spins can be detected with the CR method. Meanwhile, please change the word “discover” to

“detect” or “probe”, as V-B center in hBN is well known and is not being reported for the first time.

We thank the referee for the apt recommendation. The word “arbitrary” has been removed, and “discover” has been changed to “detect.” The new second sentence of our abstract is now:

“Here, we integrate a single nitrogen-vacancy (NV) center in diamond with scanning probe microscopy to detect, read out, and spatially map spin-based quantum sensors at the nanoscale.”

Reviewer #3:

We thank the referee for the time and effort spent reviewing and making constructive suggestions for our manuscript.

Reviewer #4:

The authors have significantly revised the manuscript, including corrections to previous statements, the addition of supplementary data, and an improved summary. Overall, the paper is much improved compared to the previous version. In particular, the analysis of the tip–sample distance is now much more convincing, and I no longer have concerns regarding the importance or validity of the main findings.

The central result of this work is the demonstration of NV center relaxometry on ensembles of VB- spin defects, enabling an estimation of the near-surface spin density. Especially with the newly added gate-dependence data, the potential of T1 mapping of boron vacancy defects as a tool for studying hBN spin defects is now much clearer. The authors have addressed all my comments to my overall satisfaction. I recommend publication in Nat. Comms.

One minor comment concerns the title. The current title, “Probing spin defects...”, is too broad and may be misleading to readers. I suggest that the authors revise the title to explicitly indicate that the work focuses on boron vacancy defects in hBN before publication.

We thank the referee for the constructive advice. We have modified our manuscript title to specify that the work focuses on boron vacancy defects in hBN:

“Probing Boron Vacancy Defects in hBN via Single Spin Relaxometry”

The manuscript by Melendez *et al.* presents experimental results of spin cross relaxation sensing of spin defects in hBN. Using shallow NV center in diamond tips as quantum sensors, they measure the spin resonance signal of V_B^- centers, including the hyperfine structures from the host ^{14}N and ^{15}N nuclear spins. In addition, they demonstrate the image of spin defects in hBN flakes at the nanoscale.

Overall, this is an interesting and practical work. The advance of the study can be identified as the experimental characterization of spin defects on the surface of 2D materials. However, the manuscript also raises some critical issues, I therefore suggest a major revision for further consideration.

Below are more specific comments:

1. The method presented in this paper is very similar to scanning NV relaxometry, which is used to study magnetic noise. For example, Finco *et al* [Nat. Commun. 12, 767 (2021)] have used scanning NV single-spin relaxometry to image non-collinear antiferromagnetic textures, and Brecht *et al* [Nano Lett. 21, 8213 (2021)] have used relaxation rates to study magnon densities. We therefore suggest the authors to explain the novelty of the current work in comparison to the previous studies.
2. I am a little confused about the introduction of the quantum network. The current experimental results have little to do with the applications of quantum networks. On the one hand, the spin coherence and optical properties of the V_B centers in hBN are far from the requirements for a quantum network. On the other hand, the observed cross-relaxation signal only means that the two spin systems are coupled, which is far from transferring a quantum state or quantum information between them.
3. NV centers with different spin relaxation time (T_1) were used for the experiment. What is the strategy for selecting NV centers to perform the cross-relaxation measurement? Meanwhile, why is the measured T_1 , even before exposure to the sample, significantly shorter than the typical T_1 of NV centers in bulk diamond?
4. In the T_1 -MR spectrum shown in Fig. 2f, the four resonance dips appear to have equal amplitudes, which is in contrast to the ratio of 1:3:3:1 observed in the ODMR spectrum. Is this consistent with theoretical expectations?
5. It would be better if you provided more details on the experimental methods and data processing, preferably in the method or SI, so that others can easily understand and reproduce the results. Critical details include the diamond tip (e.g. NV count rate, magnetic sensitivity), choice of τ in the single- τ measurements, calibration of the external magnetic field, etc.

Other minor issues

- In Fig. 1d, the horizontal axis of the T_1 measurement stops before the signal decays completely. Please indicate the fitting function used and the stretch exponent (if applicable).

- The authors write “Figure 2 were linearly baseline-corrected to isolate the contrast associated with cross-relaxation.” Please describe how this was done.
- Regarding the statement of “In addition, unlike optical methods that integrate signals over the entire sample thickness, our technique primarily probes surface defect densities...”, please provide a quantitative estimate of the depth of the sensing volume.
- The Monte Carlo simulations considered a fixed number of V_B^- spins at different densities. Could the authors clarify how the spatial scale (or cutoff) of the interacting spins was chosen and whether this choice affects the simulation results?

The manuscript titled “Probing Spin Defects via Single Spin Relaxometry” by Melendez et al. applies scanning NV magnetometry to study spin defects in naturally and isotopically purified hBN. The authors observed a decrease in the T1 time of the single NV center in the diamond tip when brought close to V_B^- spin ensembles under an external field at which the spin state transitions of NV and V_B^- match. The increase of the relaxation rate when approximate to a spin ensemble is expectable; however, the impressive aspects one is that hyperfine coupling could also be observed due to the small NV–sample distance. The second one is the T1 relaxation map of natural hBN was also able to image and quantitatively resolve V_B^- density at the nanoscale, which is crucial for optimizing spin defects in 2D materials for quantum sensing.

The observation is noteworthy, however, there remain some ambiguities in the presentation of the data and methodology in this form of the manuscript. Also, I remain uncertain about the potential significance of the results. Therefore, I cannot recommend it for publication in its current form in Nature Communications.

Major concerns are listed below:

1. The authors state that the NV–sample distance is 11.4 ± 1.5 nm, which is one of the closest NV to sample distances reported. However, this analysis was only based on a rough $B \sim 1/d$ scaling from two linecuts in lift-up mode. The magnetization and NV to sample distance could be directly obtained from linecut fitting (Thiel Science (2019) or ref. 3 in the SI), instead of using the linewidth and amplitude separately. I understand that extracting the NV to sample distance is challenging, but I was not convinced by the current analysis. Moreover, the comprehensive analysis of spin density depends on this distance, especially given that an increased NV to sample distance would lead to dramatic changes in both the central value and the range of the estimated spin density (Fig. 3e).
2. A related question, what parameters were used for the NV field in the simulations shown in Fig. S2 (e.g., NV to sample distance, magnetization density)? Why is the 80 nm curve nearly twice the value presented in Fig. S1?
3. Since the NV to sample distance normally varies significantly from tip to tip (ref. 3 in the SI), how many tips were used in this study? Can I understand that all the data, except for Fig. 2e–f, were obtained with the same tip, for which the NV height was analyzed? The NV implantation depth is 9 ± 4 nm. Is this range derived from SRIM simulations or from examining different tips?
4. Could the authors provide a comment on the spin density of the $h^{10}B^{15}N$ sample compared to the natural hBN? What is the thickness of the flake? I also did not see a strong logical connection between Fig. 2 and the other figures.

5. Another main question is what additional information the relaxation map in Fig. 3f provides. The map only shows two steps of spin density in regions with large height differences, but does not show any gradual change for different thicknesses or variations in defect density (which can be seen in the Raman map). An important point is whether the relaxation maps provide additional information compared to the Raman map (e.g., in the upper-left corner of Fig. 3b, where the Raman data show a discontinuity within a region of uniform thickness) or compared to the counts in confocal PL map (Fig. S5).
6. Is the gold substrate used to create a larger spin density contrast with respect to sample thickness? If so, how would this differ compared to a standard substrate such as SiO₂?

Some minor comments:

- a. The arrow in Fig. 1c is not described in the caption.
- b. At what field was the data for the non CR T1 curve taken? Is there any influence due to the field not being aligned with the NV center?
- c. In the main text, the average hyperfine splitting is reported as $A_{zz} = 2\pi \times (67 \pm 2)$ MHz, whereas in the caption of Fig. 2 it is given as $A_{zz} = 63$ MHz. And $A_{zz} = 65$ MHz in Fig. S17.

The revised manuscript clarifies the novelty of the work, specifically imaging the density of surface spin defects by combining the CR method with a scanning NV probe. Most of the issues raised by the reviewers have been properly addressed. I suggest a minor revision to address the following points.

1. In the second paragraph of the Introduction, the authors write, “a major limitation remains: spin-active quantum defects in two-dimensional materials that emit at telecom wavelengths...” As the CR method is unrelated to the optical properties of the target spin, I suggest removing or revising this point. Instead, the rationale for the need for nanoscale spin density imaging should be included.
2. Regarding the measurement of spin defect density, please provide a quantitative estimate of the performance of the CR method. For example, is it possible to detect a single V_B center in hBN? If not, what is the minimum detectable density? It appears that the intrinsic relaxation rate of the NV spin and the distance between the NV probe and the sample spin are important factors.
3. You mentioned the NV center (V_B center) as 3D (2D) spin sensors; in fact, they are both 0D sensors. A more accurate description is spin sensors in 3D or 2D materials.
4. In Abstract, the authors write “...with scanning probe microscopy to discover, read out, and spatially map arbitrary spin-based quantum sensors at the nanoscale.” Please remove the word “arbitrary”, as only resonant spins can be detected with the CR method. Meanwhile, please change the word “discover” to “detect” or “probe”, as V-B center in hBN is well known and is not being reported for the first time.